# Universal Biological Sequence Reranking for Improved De Novo Peptide Sequencing

**Zijie Qiu** [* 1 2] **Jiaqi Wei** [* 3 2] **Xiang Zhang** [* 4] **Sheng Xu** [1 2] **Kai Zou** [5 6] **Zhi Jin** [7 2] **Zhiqiang Gao** [2] **Nanqing Dong** [2] **Siqi Sun** [1 2]

## Abstract

De novo peptide sequencing is a critical task in proteomics. However, the performance of current deep learning-based methods is limited by the inherent complexity of mass spectrometry data and the heterogeneous distribution of noise signals, leading to data-specific biases. We present RankNovo, the first deep reranking framework that enhances de novo peptide sequencing by leveraging the complementary strengths of multiple sequencing models. RankNovo employs a listwise reranking approach, modeling candidate peptides as multiple sequence alignments and utilizing axial attention to extract informative features across candidates. Additionally, we introduce two new metrics, PMD (Peptide Mass Deviation) and RMD (Residual Mass Deviation), which offer delicate supervision by quantifying mass differences between peptides at both the sequence and residue levels. Extensive experiments demonstrate that RankNovo not only surpasses its base models used to generate training candidates for reranking pre-training, but also sets a new state-of-the-art benchmark. Moreover, RankNovo exhibits strong zero-shot generalization to unseen models—those whose generations were not exposed during training, highlighting its robustness and potential as a universal reranking framework for peptide sequencing. Our work presents a novel reranking strategy that fundamentally challenges existing single-model paradigms and advances the frontier of accurate de novo sequencing. Our source code is provided on GitHub [1].

*Equal contribution [1]Fudan University [2]Shanghai Artificial Intelligence Laboratory [3]Zhejiang University [4]University of British Columbia [5]NetMind.AI [6]ProtagoLabs Inc [7]Soochow University. Correspondence to: Siqi Sun <siqisun@fudan.edu.cn>, Nanqing Dong <dongnanqing@pjlab.org.cn>.

*Proceedings of the $42^{nd}$ International Conference on Machine Learning*, Vancouver, Canada. PMLR 267, 2025. Copyright 2025 by the author(s).

[1]https://github.com/BEAM-Labs/denovo

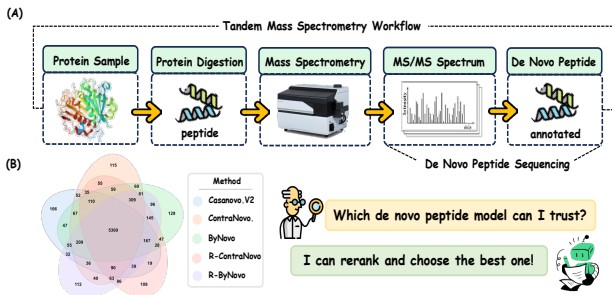

*Figure 1.* (A) **De Novo Peptide Sequencing Workflow Using Tandem Mass Spectrometry**: Our objective is to predict peptide sequences from MS/MS spectra, as illustrated in the final two steps. (B) **Motivation for RankNovo**: Current de novo peptide sequencing models exhibit data preference in their peptide predictions. RankNovo improves overall prediction accuracy by reranking the outputs of these models to identify the optimal sequence.

## 1. Introduction

Identifying proteins is a critical task in proteomics, with mass spectrometry-based shotgun proteomics being widely regarded as the predominant technique for this purpose (Aebersold & Mann, 2003). As shown in Figure 1, this process begins with the enzymatic digestion of proteins into smaller peptide fragments, which are then analyzed using tandem mass spectrometry (MS/MS) to generate spectra (Nesvizhskii et al., 2003). These spectra are subsequently interpreted to infer peptide sequences, enabling precise identification and characterization of proteins. This foundational approach is pivotal for advancing research in proteomics (Aebersold & Mann, 2003).

Proteomics utilizes two primary methodologies for peptide sequence identification: database searching (Ma et al., 2003; Chen et al., 2020; Leprevost et al., 2014; Shteynberg et al., 2011; Chi et al., 2018) and de novo sequencing (Dančík et al., 1999; Chi et al., 2013). In database searching, experimental spectra are matched against pre-existing entries in protein databases to identify the most likely sequences. Although effective for identifying known peptides, this approach is inherently constrained by the completeness of the database, posing challenges when encountering novel

or uncharacterized sequences (Karunratanakul et al., 2019; Hettich et al., 2013). On the other hand, de novo sequencing leverages the intrinsic patterns of tandem mass spectra to directly infer peptide sequences without requiring a reference database, enabling the discovery of novel peptides. Consequently, de novo sequencing has emerged as a critical technique for peptide identification, significantly advancing the scope of proteomic analysis (Ng et al., 2023).

Over the past two decades, de novo sequencing has made substantial progress, evolving from graph-theoretic and dynamic programming-based methods to more sophisticated approaches driven by deep learning (Ma et al., 2003; LeCun et al., 2015; Gao et al., 2023). DeepNovo (Tran et al., 2017) was the first to apply deep learning to de novo sequencing, which inspired a series of subsequent models(Zhou et al., 2017; Karunratanakul et al., 2019; Yang et al., 2019; Liu et al., 2023). More recently, Transformer architectures have been introduced to model de novo sequencing as a machine translation task (Yilmaz et al., 2022; Mao et al., 2023; Eloff et al., 2023a; Yang et al., 2024b; Xia et al., 2024a). Building upon this foundation, ContraNovo (Jin et al., 2024) further advanced the field by incorporating multimodal alignment strategies, achieving state-of-the-art performance.

Despite recent advancements in de novo peptide sequencing, these methods still exhibit notable accuracy limitations compared to traditional database search approaches (Muth et al., 2018). The primary challenge stems from the inherent complexity of mass spectrometry data, which consists of a mixture of heterogeneous distributions. This complexity is driven by variations in experimental conditions, such as differences in instrumentation, protocols, and target protein species, each of which introduces distinct noise patterns into the acquired spectra (Zubarev & Mann, 2007; Chang et al., 2016). As shown in Fig. 1(B), no model is exempt from issues of generalization and preferential bias, as evidenced by the presence of unique correct predictions from models that otherwise exhibit weaker overall performance. This observation motivates a rethinking of de novo peptide sequencing as a reranking task, where a trained meta-model selects the optimal prediction from a collection of outputs generated by multiple de novo models.

In this paper, we introduce RankNovo, a novel deep reranking framework designed to address the preferential bias challenges inherent in peptide sequencing. In such a complex task, peptide candidates generated for the same spectra often exhibit only minor mass differences. To effectively differentiate between these closely related candidates, RankNovo employs a list-wise reranking approach, processing and reranking all candidates in a single forward pass. This strategy enables the model to incorporate information across candidates, facilitating more precise discrimination between similar sequences. This approach stands in contrast to tra-

ditional pairwise comparison frameworks commonly used in Natural Language Processing tasks (Ouyang et al., 2022; Jiang et al., 2023). To implement this reranking strategy, RankNovo formulates peptide candidates as a Multiple Sequence Alignment (MSA) (Jumper et al., 2021; Rao et al., 2021; Abramson et al., 2024) and applies axial attention to extract sequential features. In particular, column-wise attention plays a crucial role in enabling the flow of information and intricate comparisons between candidates (Huang et al., 2019; Ho et al., 2019; Wang et al., 2020a). Additionally, spectrum features are extracted using a Transformer encoder and integrated into the peptide track via a cross-attention mechanism.

Moreover, the key concentration on amino acid masses in de novo peptide sequencing (Jin et al., 2024) inspires us to propose two novel metrics, PMD (Peptide Mass Deviation) and RMD (Residual Mass Deviation), as a more nuanced replacement of typical reranking losses such as binary classification loss. The two metrics quantitatively evaluate the mass difference between peptides at both the peptide and residue levels to provide more accurate supervision scores for RankNovo.

Experimental results show that RankNovo achieves state-of-the-art performance on de novo sequencing benchmarks, outperforming each of its component base models, including the current SOTA model, ContraNovo. We also conducted detailed analytical and ablation studies to verify the robustness of the model. Furthermore, we demonstrate that RankNovo, when trained on specific base models, can be effectively applied in a zero-shot setting to peptide predictions from unseen sequencing models, highlighting its strong transferability and its ability to capture deep knowledge for assessing peptide-spectrum matching performance.

The contributions of this paper can be summarized as follows: (1) We introduce the first deep learning-based reranking framework for peptide de novo sequencing, designed to bridge the gap between existing methods, thereby unleashing their complementary potentials. (2) We propose RankNovo, a list-wise reranking framework that models candidates as multiple sequence alignments (MSA) and uses axial attention to extract informative features. (3) We further introduce two novel metrics, PMD and RMD, for accurate measurement of mass differences between peptides, providing precise supervised signals for reranking models. (4) Extensive experiments demonstrate that RankNovo not only surpasses each of its individual ensemble components but also generalizes effectively to unseen models in a zero-shot setting, highlighting its robustness and adaptability.

## 2. Related Work

### 2.1. De Novo Peptide Sequencing

De novo sequencing algorithms have witnessed the adaption from early dynamic programming with rule-based scoring functions (Ma et al., 2003; Chi et al., 2010; Ma, 2015) to deep-learning based end-to-end regression, mainly based on transformer architecture (Yilmaz et al., 2022; Mao et al., 2023; Eloff et al., 2023a; Yang et al., 2024b; Jin et al., 2024; Zhou et al., 2024; Eloff et al., 2023b; Xia et al., 2024a; Zhang et al., 2025; Yang et al., 2024a; Xia et al., 2024b).

Despite these advancements, current methods still face inherent limitations due to the complexity of spectra. In this study, we aim to address these limitations by developing an effective reranking model to select the best matching candidate, thereby enhancing the overall capability of the de novo sequencing algorithm.

### 2.2. Candidate Reranking

In the reranking task, methods are typically categorized into three types: point-wise, pair-wise, and list-wise (Zhuang et al., 2023). The point-wise method independently evaluates the relevance of a single query-candidate pair (Nogueira et al., 2019; 2020). The pair-wise method assesses the relative relevance between two candidate pairs for a given query (Burges et al., 2005; Burges, 2010; Ouyang et al., 2022; Jiang et al., 2023). The list-wise method considers the relevance of all candidate pairs for each query collectively, utilizing all candidate features, which enhances performance potential (Han et al., 2020; Gao et al., 2021; Ren et al., 2021; Cao et al., 2007; Xia et al., 2008).

The inherent complexity of natural language processing has traditionally constrained reranking algorithms to rely predominantly on weak supervision signals, particularly relative preference indicators, irrespective of the methodological approach employed (point-wise, pair-wise, or list-wise). Our research presents a novel contribution through the development of precise metric frameworks (PMD and RMD) that facilitate effective list-wise reranking of candidate peptides. This metric-driven approach is implemented through an axial-attention-based peptide encoder architecture, which demonstrates superior capability in discriminating subtle variations among candidates, thereby enabling precise differentiation in the reranking process.

### 2.3. Axial Attention

Axial attention (Huang et al., 2019; Ho et al., 2019; Wang et al., 2020a;b; Choromanski et al., 2020) markedly reduces computational complexity while maintaining the ability to capture global context by applying the self-attention mechanism along specific axes of the input data. In the realm of protein modeling, modern deep learning methodologies frequently employ multiple sequence alignments (MSAs) (Feng & Doolittle, 1987) to harness the rich evolutionary and structural information embedded within proteins. For example, the MSA Transformer (Rao et al., 2021), a large-scale protein language model, utilizes axial attention to process extensive aligned MSA data efficiently. Likewise, the prominent protein structure and interaction prediction models AlphaFold2 (Jumper et al., 2021) and AlphaFold3 (Abramson et al., 2024) leverage axial attention to effectively model the input matrix in latent space, enabling a broad spectrum of applications in protein modeling and design.

Extending these foundational works, our research introduces the first application of the axial attention mechanism to peptide modeling. By examining the similarities and differences between candidates and spectra, our model effectively reranks these candidates. This innovative application expands the utility of axial attention, illustrating its potential in peptide modeling.

## 3. Method

### 3.1. Problem Formulation

De novo peptide sequencing seeks to deduce the amino acid sequence from a given mass spectrum. Formally, the input set $\mathcal{G} = \{\delta, m^{\text{prec}}, c^{\text{prec}}\}$ is composed of three elements: the spectrum $\delta$, a collection of mass-to-charge ratios (m/z) and intensity signals; the precursor charge $c^{\text{prec}}$, an integer; and the precursor mass $m^{\text{prec}}$, a floating-point value. A spectrum containing $\tilde{k}$ peaks (signal pairs) can be represented as $\{(\lambda_i, \tilde{\mathbf{I}}_i)\}_{i=1}^{\tilde{k}}$. The objective is to identify a set of potential residues, defined as $\mathcal{R} = \{\mathrm{r}_1, \mathrm{r}_2, \ldots, \mathrm{r}_n\}$ providing the input $\mathcal{G}$. The core concept of RankNovo is to integrate multiple relatively weak yet diverse de novo models and to train the model to select the optimal solution among their outputs. We referred to these models providing candidate predictions as base models.

### 3.2. Spectrum and Peptide Embedding

**Spectrum embedding** We filter the peaks in the spectrum $\delta_i$ with a m/z range of $[\mu_{\min}, \mu_{\max}]$. Then, the intensities $\tilde{\mathbf{I}}$ of the remaining $k$ peaks are square-root transformed and normalized as $\mathbf{I}_i = \frac{\sqrt{\tilde{\mathbf{I}}_i}}{\sum_{j=1}^{k} \sqrt{\tilde{\mathbf{I}}_j}}$.

Following previous works, we use a fixed sinusoidal embedding function $\mathbf{s}^{m/z}$ to project m/z signal into $d$-dimensional. Since all $\mu$ falls between $\mu_{min}$ and $\mu_{max}$, we embed the ratio of $\mu$ and $\mu_{min}$ and use $\frac{\mu_{max}}{\mu_{min}}$ as the scale basis of wave

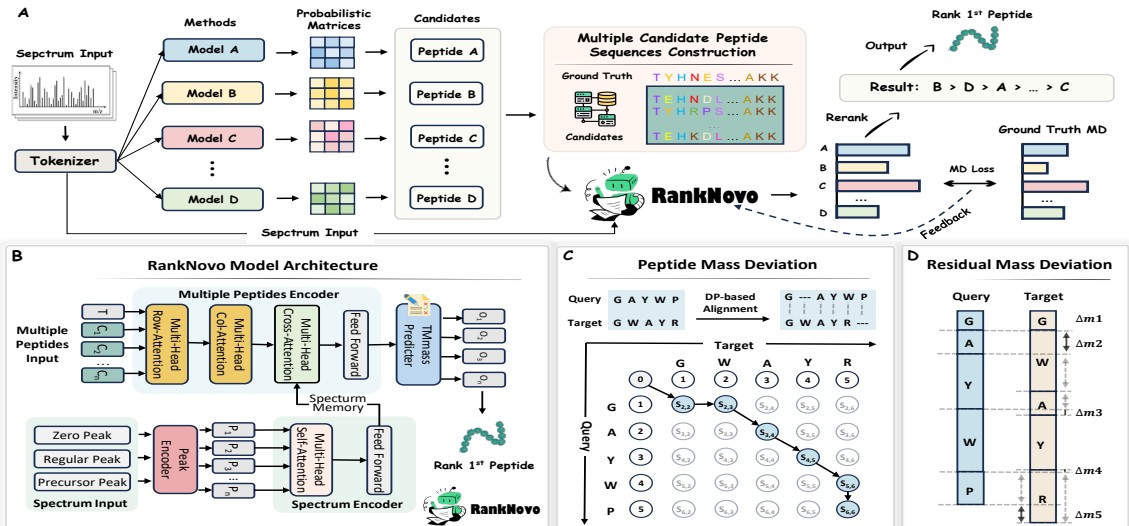

*Figure 2.* An overview of the RankNovo architecture. (A) Multiple base models generate peptide sequence candidates from the spectrum input, which are subsequently reranked by RankNovo. (B) The architecture of RankNovo incorporates a multi-peptide encoder, utilizing axial attention along both row and column dimensions and cross-attention to effectively integrate spectrum features. (C) The coarse-grained PMD metric assesses peptide-level differences through dynamic programming-based sequence alignment. (D) The fine-grained RMD metric provides a more granular assessment by capturing residue-level mass deviations between the query and target peptides.

length:

$$\mathbf{s}^{m/z}(\mu, i) = \begin{cases} \sin((2\pi\frac{\mu}{\mu_{\min}})/((\frac{\mu_{\max}}{\mu_{\min}})^{\frac{k}{d}})), & \text{if } i = 2k \\ \cos((2\pi\frac{\mu}{\mu_{\min}})/((\frac{\mu_{\max}}{\mu_{\min}})^{\frac{k}{d}})), & \text{if } i = 2k+1 \end{cases} \quad (1)$$

Intensity signals are projected to $d$ dimension with a linear layer because of its relatively lower accuracy and are summed with corresponding m/z vectors as the initial spectrum embedding $\mathbf{E^0}$, with shape $[k, d]$. Here, no additional positional embeddings are used, as the peaks are inherently unordered in nature.

**Peptide Candidate Embedding** De novo sequencing is a mass-centric task. Therefore, the prefix and suffix masses of a residual are also embedded in addition to a learnable amino acid embedding, following ContraNovo. Given the model dimension $d$, the dimensions of the learnable embedding, prefix, and suffix, denoted as $d_{\text{res}}, d_{\text{prefix}}, d_{\text{suffix}}$, are set to $\frac{d}{2}, \frac{d}{4}$ and $\frac{d}{4}$ respectively. Moreover, the precursor of spectrum are embedded into $d_{\text{prec}}$ ($\frac{d}{2}$) dimensions as the sum of $d_{\text{prec}}$-dimensional precursor mass vector and precursor charge vector (Jin et al., 2023). All masses $m$ are embedded using fixed sinusoidal positional embedding:

$$\mathbf{H}^{\mathcal{T}}(m, i) = \begin{cases} \sin\left(\frac{2\pi m}{10000^{k/d}\tau}\right), & \text{for } i = 2k \\ \cos\left(\frac{2\pi m}{10000^{k/d}\tau}\right), & \text{for } i = 2k+1 \end{cases}, \quad \mathcal{T} \in \{\text{prefix, suffix, prec}\} \quad (2)$$

While the learnable embedding functions for amino acids and precursor charges can be represented as $\mathbf{B}^{\text{res}}$ and $\mathbf{B}^{\text{charge}}$. Then the initial peptide embedding $\mathbf{h}^0 =$

$[\mathbf{h}_{\text{cls}}, \mathbf{h}_1^0, \mathbf{h}_2^0, \dots, \mathbf{h}_\ell^0]$ can be denoted as:

$$\mathbf{h}_i^0 = \mathbf{B}^{\text{res}}(\text{res}_i) \oplus \mathbf{H}^{\text{prefix}}(m_i^{\text{prefix}}) \oplus \mathbf{H}^{\text{suffix}}(m_i^{\text{suffix}})$$
$$\mathbf{h}_{\text{cls}} = \mathbf{B}^{\text{res}}(\text{cls}) \oplus [\mathbf{H}^{\text{prec}}(m^{\text{prec}}) + \mathbf{B}^{\text{charge}}(c^{\text{prec}})] \oplus \mathbf{0}_{d/2} \quad (3)$$

where $\oplus$ denotes the concatenation operation over the sequence length dimension and $\mathbf{0}_{d/2}$ denotes learnable 0-quality embedding. RankNovo processes all peptide candidates in a single forward pass. Thus, each candidate's embedding is padded to the longest and stacked into the initial MSA embedding $\mathbf{S}^0$. Additionally, a learnable positional embedding is added to each row of $\mathbf{S}^0$, ensuring the model is aware of the token order in a peptide. It is important to note that shuffling the order of rows does not affect the loss prediction due to the absence of column-wise positional embedding.

### 3.3. Accurate Assessment of Peptide Difference

In the context of peptide sequencing tasks, accurate labeling of predictions is crucial for meta-models to effectively identify optimal predictions. Conventional labeling methods for base model predictions, such as binary classification of correctness or edit distance metrics, inadequately capture the nuanced differences between predicted and actual peptide sequences, particularly with respect to amino acid masses, which are fundamental to the mass-centric sequencing process.

To address these limitations, we introduce PMD and RMD, two novel metrics designed for precise quantification of

mass differences between peptides. These metrics provide more informative and accurate supervision for RankNovo to learn from, thereby improving its discriminative capabilities. PMD and RMD are complementary training objectives during training.

**Peptide-level Assessment (PMD)** PMD employs a dynamic programming approach analogous to the Needleman-Wunsch algorithm (Needleman & Wunsch, 1970) for sequence alignment, with a specific focus on amino acid masses. Given a set of all possible residues, including both amino acids and post-translational modifications (PTMs), defined as $\mathcal{R} = \{r_1, r_2, \ldots, r_n\}$, where $n$ denotes the number of distinct residue types, we introduce a corresponding mass look-up table $\mathcal{M} : \mathcal{R} \rightarrow \mathbb{R}^+$. Here, $\mathcal{M}(r_i)$ represents the mass of residue $r_i$ for $i \in \{1, 2, ..., n\}$. We define the divergence score matrix $\mathbf{P} \in R^{n \times n}$, where

$$\mathbf{P}_{i,j} = \begin{cases} 0, & \text{if } i = j \\ |\mathcal{M}(r_i) - \mathcal{M}(r_j)|, & \text{if } i \neq j \end{cases}, \quad i, j \in \{1, 2, \ldots, n\} \tag{4}$$

The gap penalty $\mathbf{g}$ is formalized as the expected symmetric divergence between two distinct residues, given by:

$$\mathbf{g} = \mathbb{E}_{i \neq j} \left[ \mathbf{P}(\mathbf{r}_i, \mathbf{r}_j) \right] = \frac{1}{n(n-1)} \sum_{i=1}^{n} \sum_{\substack{j=1 \\ j \neq i}}^{n} |\mathcal{M}(\mathbf{r}_i) - \mathcal{M}(\mathbf{r}_j)| \tag{5}$$

where $\mathbb{E}_{i \neq j} \left[ \mathbf{P}(\mathbf{r}_i, \mathbf{r}_j) \right]$ is the expectation of the symmetric divergence between the residues.

Given a query peptide sequence $\mathbb{Q} = [r_{q_1}, r_{q_2}, \ldots, r_{q_n}]$ and a target peptide sequence $\mathbb{K} = [r_{k_1}, r_{k_2}, \ldots, r_{k_m}]$, where $n$ and $m$ represent the lengths of the predicted and correct peptides respectively, we initialize a matrix $\mathbf{F} \in R^{(n+1) \times (m+1)}$. The matrix $\mathbf{F}$ is populated using the following recurrence relation:

$$\mathbf{F}_{i,j} = \begin{cases} 0, & \text{if } i = 1, j = 1 \\ \mathbf{g}(i-1), & \text{if } i \neq 1, j = 1 \\ \mathbf{g}(j-1), & \text{if } i = 1, j \neq 1 \\ \min \left\{ \mathbf{F}_{i-1,j-1} + \mathbf{P}_{q_{i-1},k_{j-1}}, \mathbf{F}_{i-1,j} + \mathbf{g}, \mathbf{F}_{i,j-1} + \mathbf{g} \right\}, & \text{otherwise} \end{cases}$$
$$i \in \{1, 2, \ldots, n+1\}, \quad j \in \{1, 2, \ldots, m+1\} \tag{6}$$

The final output of PMD between the two peptides is computed as $\mathbf{F}_{n+1,m+1}/\mathbf{g}$. Dividing by $\mathbf{g}$ normalizes the value to an order of magnitude around $10^0$, facilitating model fitting. PMD achieves a score of zero only when the predicted peptide exactly matches the correct peptide, making it a precise metric for peptide distance assessment in mass spectrometry-based proteomics.

**Residual-level Assessment (RMD)** In addition to the peptide-level metric PMD, which the meta-model uses to select the top prediction, we introduce a more fine-grained peptide difference score, RMD. This metric takes advantage

of the intrinsic properties of mass spectrometry data. In mass spectrometry, peptide bonds between amino acids are cleaved, generating b- and y-ions. The b-ions, originating from the N-terminus, offer a detailed structural fingerprint of the peptide.

RMD is derived from the prefix masses of the query peptide $\mathbb{Q}$ and the target peptide $\mathbb{K}$, denoted as $\widetilde{\mathbb{Q}} = [\overline{m}_{q_1}, \overline{m}_{q_2}, \ldots, \overline{m}_{q_n}]$ and $\widetilde{\mathbb{K}} = [\overline{m}_{k_1}, \overline{m}_{k_2}, \ldots, \overline{m}_{k_m}]$, where $\overline{m}_{q_i} = \sum_{j=1}^{i} \mathcal{M}(r_{q_j})$ and $\overline{m}_{k_i} = \sum_{j=1}^{i} \mathcal{M}(r_{k_j})$. This representation is closely aligned with the b-ion mass spectrum. The RMD between these two sequences is represented as a vector $\mathbf{V}$ with $n$ elements, where each element is defined as:

$$\mathbf{V}_i = \overline{m}_{q_i} - \overline{m}_{k_{\pi(i)}}, \quad \text{where } \pi(i) = \arg\min_{\tilde{j}} \left| \overline{m}_{q_i} - \overline{m}_{k_{\tilde{j}}} \right|. \tag{7}$$

Here $\pi(i)$ is a learned alignment function that seeks to minimize the mass difference between the $\mathbb{Q}$ and $\mathbb{K}$. By training the model to predict RMD, we encourage it to capture and distinguish subtle structural deviations between peptides. This residual-level task improves the model's ability to identify fine-grained peptide differences, complementing the higher-level insights given by PMD.

### 3.4. Backbone of RankNovo

The backbone of RankNovo needs to fulfill three tasks: (1) Extracting spectrum feature, (2) Extracting peptide features within and among candidates, (3) Mixing spectrum feature and peptide feature to score and rerank peptide candidates.

Spectrum feature extraction can be easily accomplished by a Transformer encoder. After embedding, the initial spectrum representation $\mathbf{E}^0$ is updated by $N_{\text{layer}}$ a repetitive self-attention layer:

$$\mathbf{E}^{(i)} = \mathcal{A}_{\text{self}}(\mathbf{E}^{(i-1)}), i = 1, 2, \ldots, N_{\text{layer}} \tag{8}$$

On the other hand, a hybrid peptide track is designed to address tasks (2) and (3) jointly. The peptide track processes the embedded multiple sequence alignment (MSA) feature $\mathbf{S}^0 \in \mathbb{R}^{c \times \ell \times d}$, where $c$ represents the number of candidates, $\ell$ is the sequence length, and $d$ is the model dimension. The final spectrum feature $\mathbf{E}^{N_{\text{layer}}} \in \mathbb{R}^{k \times d}$ is broadcasted across candidates by repeating it to shape $[c, k, d]$, and then integrated into the peptide track. The feature update mechanism is defined as:

$$\mathbf{S}^{(i)} = \mathcal{A}_{\text{cross}} \Big( \mathcal{A}_{\text{col}} \big( \mathcal{A}_{\text{row}}(\mathbf{S}^{(i-1)}) \big), \mathbf{E}^{N_{\text{layer}}} \Big) \tag{9}$$

where $\mathcal{A}_{\text{row}}$, $\mathcal{A}_{\text{col}}$, and $\mathcal{A}_{\text{cross}}$ denote the row-wise, column-wise, and cross-attention mechanisms, respectively. Here, axial attention is employed to extract peptide features and facilitate information flow between candidate peptides. The iterative application of row and column attention ensures a

receptive field that spans the entire $\ell \times k$ token grid while maintaining a reduced complexity of $\mathcal{O}(c\ell^2 + k^2\ell)$, in contrast to the $\mathcal{O}(c^2\ell^2)$ complexity of standard multi-head self-attention mechanisms. Cross-attention is integrated to incorporate spectrum features into the peptide track, allowing for enhanced alignment between peptides and spectra, and improving overall task performance.

### 3.5. Training with Joint Loss

The final MSA feature $\mathbf{S}^{N_{\text{layer}}}$ is utilized to predict the PMD and RMD between each candidate peptide and the label peptide. For the peptide-level metric, PMD, the CLS token of each candidate is extracted and passed through a linear layer to predict PMD, formulated as: PMD = Linear$(\mathbf{h}_{\text{cls}}) \in R$. Similarly, the $d$-dimensional representation of each amino acid is projected through a linear transformation to predict the residue-level RMD, expressed as: RMD = $\{\text{Linear}\left(\mathbf{h}_i^{N_{\text{layer}}}\right) \in R\}_{i=1,\dots,\ell}$. Both $\mathcal{L}_{\text{PMD}}$ and $\mathcal{L}_{\text{RMD}}$ are computed using RMSE loss. The optimization objective for training RankNovo is defined as:

$$\mathcal{L} = \lambda\mathcal{L}_{\text{PMD}} + (1 - \lambda)\mathcal{L}_{\text{RMD}} \tag{10}$$

In this work, $\lambda$ is set 0.5 consistently.

## 4. Experiments

### 4.1. Experiment Setup

**Datasets.** Following the precedent set by recent studies (Yilmaz et al., 2023; Zhang et al., 2024), we employ three public peptide-spectrum match (PSMs) datasets: MassIVE-KB (Wang et al., 2018) for training, and 9-species-V1 (Tran et al., 2017) and 9-species-V2 (Yilmaz et al., 2023) for evaluation, enabling comparisons with state-of-the-art de novo peptide sequencing methods. Detailed dataset information is provided in the Appendix A.

**Implementation Details.** RankNovo incorporates six de novo sequencing models, each varying in methodology, as base models during training. These models include Casanovo-V2, ContraNovo, ByNovo, R-Casanovo, R-ContraNovo, and R-ByNovo. Of these, Casanovo-V2 and ContraNovo are directly adopted from the original works and represent both the current and previous state-of-the-art approaches. The latter four models, ByNovo, R-ContraNovo, and R-ByNovo, are developed and trained by us. The details of base models, training settings, and hyperparameters of RankNovo can be found in Appendix B.

**Metrics.** Since the reranking task only concerns peptide-level selection, the widely accepted metric, peptide recall, is our most important metric. Peptide recall is defined as $N_{\text{match}}^{pep}/N_{\text{all}}^{pep}$, here $N_{\text{match}}^{pep}$ is the number of matched peptides and $N_{\text{all}}^{pep}$ is the number of total peptides. The identified peptide is regarded as matched to the label peptide only if every residue is matched. Here, residual matching means (1) differing by $< 0.1$ Da in mass and (2) both the prefix and suffix differing within 0.5 Da. Also, since previous works evaluate model capabilities at residual-level as well, amino acid precision is also taken into consideration. Here, amino acid precision is defined as $N_{\text{match}}^a/N_{\text{all}}^a$, meaning the percentage of matched residuals among all residuals.

### 4.2. Main Results

**Performance on 9-species-V1 Benchmark Dataset.** In our evaluation, RankNovo exhibits superior performance across all species on the pivotal benchmark, 9-species-V1, both at the peptide and amino acid levels (Table 1). Specifically, RankNovo achieves an average peptide recall of 0.660, surpassing its strongest base model, ByNovo, by 6.1%, and outperforming the current state-of-the-art, ContraNovo, by 4.3%. At the amino acid level, RankNovo reaches a precision of 0.829, outperforming ByNovo by 2.6% and ContraNovo by 4.1%. These results underscore RankNovo's ability to accurately sequence peptides and amino acids across diverse species.

Two key conclusions can be drawn from these results: first, RankNovo establishes a new state-of-the-art in de novo peptide sequencing, surpassing the previous benchmark set by ContraNovo; and second, RankNovo consistently outperforms all of its constituent base models, demonstrating its ability to effectively integrate diverse model outputs, leverage their respective strengths, and mitigate individual weaknesses, thereby reducing generalization error.

**Performance on 9-species-V2 Benchmark Dataset.** The experimental results in Table 1 clearly indicate that RankNovo consistently outperforms both baseline and comparative models on the 9-species-V2 dataset, demonstrating superior performance in amino acid precision and peptide recall. Specifically, RankNovo achieves the highest average amino acid precision of 0.906 across all species, with substantial improvements in species such as Bacillus, C. bacteria, and Honeybee. Furthermore, RankNovo attains an average peptide recall of 0.781, outperforming other models across the majority of species, with particularly strong performance in Yeast, Rice bean, and Tomato. These results emphasize the adaptability and effectiveness of RankNovo across a diverse set of species.

### 4.3. Detailed Analyses

We report average performance on the 9-species-V1 benchmark, with detailed per-species performance in Appendix E.

**Analysis of Zero-shot Performance.** We demonstrate the zero-shot capability of RankNovo by training it exclusively on predictions from the two lowest-performing base models and progressively incorporating predictions from unseen

| | Methods | *Bacillus* | *C. bacteria* | *Honeybee* | *Human* | *M.mazei* | *Mouse* | *Rice bean* | *Tomato* | *Yeast* | **Average** |
|---|---|---|---|---|---|---|---|---|---|---|---|
| | | | | | **9-species-V1** | | | | | | |
| **Amino Acid Precision** | PEAKS | 0.719 | 0.586 | 0.633 | 0.639 | 0.673 | 0.600 | 0.644 | 0.728 | 0.748 | 0.663 |
| | DeepNovo | 0.742 | 0.602 | 0.630 | 0.610 | 0.694 | 0.623 | 0.679 | 0.731 | 0.750 | 0.673 |
| | PointNovo | 0.768 | 0.589 | 0.644 | 0.606 | 0.712 | 0.626 | 0.730 | 0.733 | 0.779 | 0.687 |
| | Casanovo | 0.749 | 0.603 | 0.629 | 0.586 | 0.679 | 0.689 | 0.668 | 0.721 | 0.684 | 0.667 |
| | Casanovo V2[†] | 0.806 | 0.685 | 0.727 | 0.690 | 0.774 | 0.768 | 0.769 | 0.799 | 0.762 | 0.753 |
| | ContraNovo[†] | 0.828 | 0.706 | 0.761 | 0.771 | 0.798 | 0.799 | 0.804 | 0.808 | 0.782 | 0.784 |
| | ByNovo[†⋆] | 0.858 | 0.723 | 0.791 | 0.767 | 0.823 | 0.803 | 0.836 | 0.828 | 0.804 | 0.804 |
| | **RankNovo** | **0.874** | **0.746** | **0.810** | **0.802** | **0.840** | **0.828** | **0.859** | **0.844** | **0.816** | **0.824** |
| **Peptide Recall** | PEAKS | 0.387 | 0.203 | 0.287 | 0.277 | 0.356 | 0.197 | 0.362 | 0.403 | 0.428 | 0.322 |
| | DeepNovo | 0.449 | 0.253 | 0.330 | 0.293 | 0.422 | 0.286 | 0.436 | 0.454 | 0.462 | 0.376 |
| | PointNovo | 0.518 | 0.298 | 0.396 | 0.351 | 0.478 | 0.355 | 0.511 | 0.513 | 0.534 | 0.439 |
| | Casanovo | 0.537 | 0.330 | 0.406 | 0.341 | 0.478 | 0.426 | 0.506 | 0.521 | 0.490 | 0.448 |
| | Casanovo V2[†] | 0.646 | 0.460 | 0.527 | 0.492 | 0.592 | 0.493 | 0.628 | 0.637 | 0.629 | 0.567 |
| | ContraNovo[†] | 0.684 | 0.487 | 0.576 | 0.624 | 0.628 | 0.563 | 0.676 | 0.655 | 0.669 | 0.618 |
| | ByNovo[†⋆] | 0.708 | 0.499 | 0.597 | 0.584 | 0.639 | 0.545 | 0.696 | 0.667 | 0.676 | 0.623 |
| | **RankNovo** | **0.738** | **0.539** | **0.630** | **0.642** | **0.672** | **0.583** | **0.733** | **0.691** | **0.703** | **0.660** |
| | | | | | **9-species-V2** | | | | | | |
| **Amino Acid Precision** | Casanovo V2[†] | 0.888 | 0.791 | 0.823 | 0.872 | 0.877 | 0.813 | 0.891 | 0.891 | 0.915 | 0.862 |
| | ContraNovo[†] | 0.901 | 0.807 | 0.848 | 0.920 | 0.896 | 0.839 | 0.913 | 0.898 | 0.919 | 0.882 |
| | ByNovo[†⋆] | 0.920 | 0.823 | 0.876 | 0.917 | 0.914 | 0.841 | 0.932 | 0.912 | 0.934 | 0.897 |
| | **RankNovo** | **0.926** | **0.838** | **0.885** | **0.929** | **0.920** | **0.860** | **0.938** | **0.918** | **0.938** | **0.906** |
| **Peptide Recall** | Casanovo V2[†] | 0.793 | 0.558 | 0.669 | 0.712 | 0.754 | 0.555 | 0.772 | 0.783 | 0.837 | 0.714 |
| | ContraNovo[†] | 0.815 | 0.575 | 0.711 | 0.820 | 0.780 | 0.616 | 0.799 | 0.794 | 0.854 | 0.752 |
| | ByNovo[†⋆] | 0.833 | 0.582 | 0.731 | 0.789 | 0.799 | 0.596 | 0.814 | 0.807 | 0.871 | 0.758 |
| | **RankNovo** | **0.851** | **0.620** | **0.752** | **0.820** | **0.813** | **0.629** | **0.836** | **0.822** | **0.885** | **0.781** |

*Table 1.* Evaluation of RankNovo in comparison to baseline and base models on the 9-species-V1 and 9-species-V2 test sets. Bolded entries indicate the best-performing models. The symbol "†" indicates that the model serves as both a baseline and a base model. "⋆" signifies that the base models were developed and trained by us. Here, ByNovo is the best base model. So the performance of three self-trained base models (R-Casa, R-Contra, R-By) is only provided in Appendix D.1.

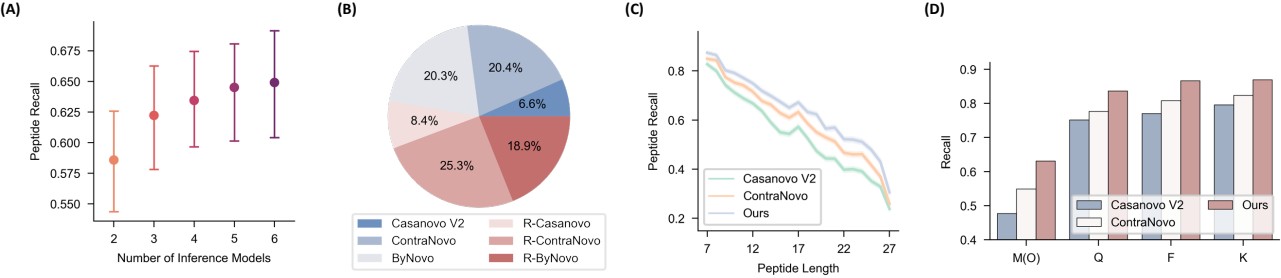

*Figure 3.* (A) Zero-shot performance of RankNovo when trained on two base models. (B) Unique-correctly selected percentage of base models. (C) Influence of peptide length. (D) The performance comparison of amino acids with similar masses.

models into the candidate sets during inference (Figure 3 (A)). The detailed experimental setup is provided in Appendix E.1. As the number of inference models increases, the average peptide recall improves, rising from 0.586 with 2 models to 0.649 with 6 models. These results highlight the robust zero-shot capability of RankNovo, demonstrating its efficiency in reranking predictions from models not used during training. This underscores the value of RankNovo for future applications in de novo peptide sequencing.

**Contribution of Each Base Model.** Given the varying capabilities of base models, it is crucial to ensure that each contributes meaningfully to RankNovo's performance. Otherwise, their predictions may introduce unnecessary noise during the reranking process. We analyzed peptide candidates' and RankNovo's selections using the Bacillus species data from the 9-species-V1 benchmark. We filtered spec-

trum samples to retain those where (1) RankNovo's chosen peptide matched the labeled peptide and (2) RankNovo's choice was provided by only one base model. For these filtered samples, we calculated the percentage of times each base model was chosen, using this as a measure of contribution. As illustrated in Figure 3 (B), Casanovo-V2 had the lowest contribution at 6%, while R-ByNovo had the highest at 30%. These results demonstrate that every model contributes to the final performance, as removing any of them would lead to failures on specific test samples.

**Analysis of Peptide Length.** We assess the performance of RankNovo and baselines in recognizing peptides of varying lengths, with a particular emphasis on their effectiveness for both shorter and longer peptides. As shown in Figure 3 (C)), our findings reveal that RankNovo exhibits significantly higher recall compared to ContraNovo for shorter peptides,

| Objective | Avg. Pep. Recall |
|-----------|------------------|
| Point-wise | 0.647 |
| Pair-wise | 0.648 |
| List-wise | 0.646 |
| PMD+RMD | **0.660** |

*Table 2.* Average Peptide Recall on 9-species-V1 test set under the training objective of different reranking losses.

| ID | PMD | RMD | Col-Attn. | Avg. Pep. Recall |
|----|-----|-----|-----------|------------------|
| 1 | | ✔ | ✔ | 0.650 |
| 2 | ✔ | | ✔ | 0.652 |
| 3 | ✔ | ✔ | | 0.653 |
| 4 | ✔ | ✔ | ✔ | **0.660** |

*Table 3.* Ablation of training metrics combination and column-wise attention modules.

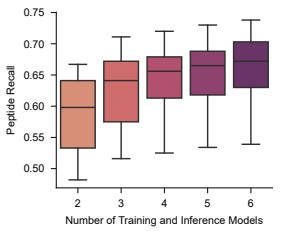

*Table 4.* Ablation study of base model combinations.

suggesting enhanced proficiency in recognizing these sequences. As we analyze longer peptides, a discernible trend emerges: the recall for both models shows a downward trajectory, indicative of a decline in recognition capability as peptide length increases. This reduction in performance can be attributed to the heightened complexity associated with longer peptide structures, which may impede model accuracy. Nevertheless, RankNovo consistently outperforms ContraNovo, although the margin of superiority narrows with increasing peptide length.

**Analysis of Amino Acids with Similar Masses.** In de novo peptide sequencing, a sequence is deemed accurately reconstructed only when each residue in the predicted peptide aligns with its corresponding residue in the reference sequence. Prediction accuracy varies across different amino acids, particularly for those with similar masses, which are challenging to distinguish due to nearly overlapping spectral profiles. For instance, oxidized methionine (M(O)) and phenylalanine (F) differ by 0.33 Da, while lysine (K) and glutamine (Q) differ by 0.46 Da. Figure 3 (D))compares RankNovo with two baseline models, Casanovo-V2 and ContraNovo. Utilizing the 9-species-V1 dataset, recall was computed for each amino acid. Notably, RankNovo achieves an 8.0% improvement in recall for M(O) relative to the baseline models. These results underscore RankNovo's enhanced ability to differentiate between amino acids with closely related masses, effectively capturing subtle mass variations within peptide sequences.

### 4.4. Reranking Framework Comparison and Ablation Study

We compare the performance of RankNovo with other reranking frameworks and conduct ablation studies on its key components. The evaluation is performed on the 9-species-V1 benchmark. Detailed results and analysis are presented in Appendix C and D.

**Reranking Framework Comparison.** We compare RankNovo with three types of reranking frameworks outlined in RankT5 (Zhuang et al., 2023): point-wise, pair-wise, and list-wise reranking. Following RankT5's methodology, these frameworks are implemented using identical backbone models, differing only in their training objectives. Detailed

descriptions are provided in Appendix C. As shown in Table 2, the three frameworks exhibit comparable performance when reranking de novo sequencing results, achieving approximately 0.647 peptide recall. However, this falls significantly short of the 0.660 recall achieved by RankNovo using our novel metrics, PMD and RMD. These results underscore the specialized efficacy of the RankNovo in peptide sequencing tasks.

**Base Model Combinations Ablation.** We conducted ablation studies to assess the impact of using fewer base models on overall performance. We created five subsets of the final base model set, each containing a different number of base models, and compared the performance of RankNovo when trained and tested with the outputs of these model sets. The selection criteria for these subsets are detailed in Appendix D.2. The results (Figure 4) demonstrate a consistent increase in peptide recall as the number of base models increases. This observation supports the hypothesis that a greater diversity of choices leads to improved performance.

**Training Objective Ablation.** Two novel metrics, PMD and RMD, provide the learning objective for RankNovo. The results of experiments 1, 2, and 4 in Table 3 demonstrate that the absence of either metric leads to a decrease in peptide recall. The combination of both metrics is necessary to achieve optimal performance.

**Backbone Model Ablation.** The results of experiments 3 and 4 in Table 3 reveal a decline in performance without the column-wise attention module, as evidenced by the 0.653 peptide recall after its removal, compared to the original 0.660. This finding supports the hypothesis that incorporating axial attention facilitates the integration of peptide features and contributes to optimal performance.

## 5. Conclusion

In this paper, we introduced RankNovo, a novel list-wise deep reranking framework designed to enhance the accuracy of de novo peptide sequencing under the guidance of our mass deviation metrics, PMD and RMD. RankNovo achieves new state-of-the-art performance on established benchmarks and exhibits strong zero-shot generalization capabilities. The primary limitation of RankNovo lies in the

relatively lower inference speed due to the proportional time cost in collecting peptide candidates (Appendix E.5). Future work could explore efficient candidate sampling methods, such as utilizing base models with partially shared weights to reduce computational overhead.

Despite the speed constraints, RankNovo represents the first deep reranking framework to offer a flexible trade-off between inference time and performance, introducing a novel perspective for performance enhancement. We anticipate that, influenced by RankNovo, future algorithms in this field will benefit from the synergistic approach of simultaneously improving single-model performance and developing advanced reranking strategies.

## Impact Statement

Our work advances machine learning applications in proteomics by developing a novel deep learning framework for peptide de novo sequencing from tandem mass spectrometry data. By improving sequencing accuracy through seq-to-seq architecture with reranking optimization, this method could accelerate biomedical discoveries in disease biomarker identification and personalized medicine development.

We foresee two primary societal impacts: (1) Enabling more accessible protein analysis that could lower costs for clinical proteomics applications, and (2) Facilitating drug discovery through improved characterization of therapeutic peptides. Potential ethical considerations include ensuring equitable access to this technology across research communities and responsible handling of sensitive health data in clinical applications.

While there are no immediate malicious use cases inherent to the algorithm itself, we acknowledge that any proteomics advancement carries dual-use potential. To mitigate risks, we will open-source our model with usage guidelines and collaborate with domain experts to establish best practices for clinical translation. This work ultimately aims to strengthen the positive impact of machine learning in driving scientific discovery while maintaining responsible innovation principles.

## Acknowledgement

This project was partially supported by Shanghai Artificial Intelligence Laboratory (S.S.). This work is partially supported by Netmind.AI and ProtagoLabs Inc. This work is also partially supported by CURE (Hui-Chun Chin and Tsung-Dao Lee Chinese Undergraduate Research Endowment) (24924), and the National Undergraduate Training Program on Innovation and Entrepreneurship grant(24924).

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

# A. Dataset Details

MassIVE-KB is a widely used training dataset employed in previous studies such as GLEAMS (Bittremieux et al., 2022) and CasaNovo. Its popularity stems from its substantial size, comprising 30 million PSMs, and its diverse distribution, characterized by sources from different instruments and a rich variety of posttranslational modifications. The 9-species-V1 dataset, introduced by DeepNovo, contains approximately 1.5 million PSMs with a database search false discovery rate of 1%. These PSMs are derived from nine distinct experiments conducted on the same instrument but analyzing peptides from various species, ensuring the dataset's diversity. The 9-species-V2 dataset, an updated version of 9-species-V1 collected in CasaNovo-V2, contains 2.8 million PSMs. Building upon V1, V2 was refined using the Crux protein identification tool (McIlwain et al., 2014) and filtered with a Percolator (Spivak et al., 2009) q-value < 0.01, enhancing its quality.

In addition to the PSMs datasets, we collected the best peptide predictions from six baseline models: Casanovo, ContraNovo, ByNovo, Re-Casanovo, Re-ContraNovo, and Re-ByNovo for each spectrum. Due to the substantial size of the MassIVE-KB training dataset and the computational constraints of beam search, we employed greedy decoding for peptide collection in the training phase. Additionally, spectrums that are correctly predicted by all six base models are excluded, leaving 7 million spectrums for the training set. Conversely, for the evaluation datasets (9-species-V1 and 9-species-V2), we utilized beam search decoding with a beam size of 5. This approach aligns with previous works and enables optimal benchmark performance during evaluation.

# B. Implementation Details

### B.1. Hyperparameters

RankNovo is implemented with the following hyperparameters: 8 layers for both the spectrum encoder and peptide feature mixer, 8 attention heads, a model dimension of 512, a feed-forward dimension of 1024, and a dropout rate of 0.30.

For spectrum and peptide preprocessing, spectra are filtered according to the following criteria: minimum m/z ratio of 50.5 Da, maximum m/z ratio of 4500.0 Da, maximum peak number of 300, precursor m/z tolerance of 2.0 Da, and precursor mass tolerance of 50 ppm. Spectra with more than 300 peaks are truncated, retaining only the 300 peaks with the highest intensities. Spectra that do not satisfy the precursor m/z tolerance and precursor mass tolerance are removed. Additionally, peptides longer than 100 amino acids are truncated. During evaluation, all base models generate peptides using a beam search with a size of 5.

RankNovo is trained using an AdamW optimizer with a learning rate of 1e-4 and weight decay of 8e-5. The model is trained with a batch size of 256 for 5 epochs, including a 1-epoch warm-up period. A cosine learning rate scheduler is employed, and gradients are clipped to 1.5 using L2 norm. The training is conducted on 4 A100 40G GPUs.

### B.2. Baselines

Our benchmark evaluation first compares RankNovo with its base model components to assess the effectiveness of the reranking framework. The components include Casanovo-V2, ContraNovo, ByNovo, R-Casanovo, R-ContraNovo, and R-ByNovo, which collectively represent both current and previous state-of-the-art models for de novo sequencing, particularly ContraNovo and Casanovo-V2. For consistency with prior work, we also evaluate four additional benchmark algorithms: DeepNovo, PointNovo, Casanovo-V1, and PEAKS. Notably, PEAKS employs a dynamic programming-based approach, while the remaining three are deep learning-based models.

### B.3. Base Model Selection

The selection of base models decides the performance upper bound of RankNovo. The selection of base models should follow three criteria: (1) The training datasets of each base model should have no interest with the test dataset, which is a common data leakage problem in ensemble learning. (2) Base models should be diverse in data preference.

In our research, we selected six models for de novo peptide sequencing as our base models: Casanovo (Yilmaz et al., 2022; 2023), ContraNovo (Jin et al., 2024), ByNovo, R-Casanovo, R-ContraNovo, and R-ByNovo as shown in Figure 4. All these models are Transformer-based, but each employs different methodologies. Casanovo and ContraNovo are based on previous work, and we directly use the official checkpoints for these models. The latter four models, ByNovo, R-ContraNovo, and R-ByNovo, are developed and trained by us:

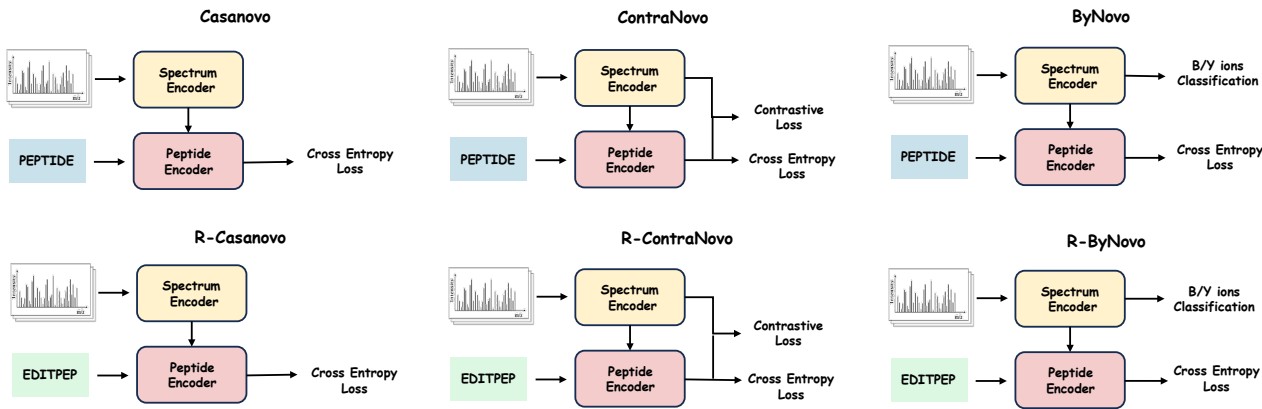

*Figure 4.* The architecture of six base models for de novo peptide sequencing.

1. **CasaNovo V2**: This fundamental Transformer-based de novo sequencing model treats de novo sequencing as a sequence-to-sequence machine translation task, translating spectra into peptides.

2. **ContraNovo**: This model leverages contrastive learning to enhance feature extraction. It also more effectively utilizes amino acid masses, as well as prefix and suffix masses, during the encoding and decoding processes.

3. **ByNovo**: Developed in-house to increase model diversity, ByNovo incorporates the prediction of BY ions using the output of the spectrum encoder as an auxiliary task. Refer to Appendix B.4 for ByNovo implementation details.

4. **R-Casanovo**: Inspired by recent studies (Eloff et al., 2023a; Wu et al., 2023), this model trains to decode peptide sequences in reverse. The one-way nature of auto-regressive decoding leads to different results when the sequence is decoded in reverse. R-Casanovo is the reverse-decoding version of Casanovo.

5. **R-ContraNovo**: The reverse decoding version of ContraNovo.

6. **R-ByNovo**: The reverse decoding version of ByNovo.

To avoid the data leakage problem, these six models are all trained on Massive-KB, the largest available de novo peptide sequencing dataset in public, and are evaluated in a zero-shot manner on the Nine-Species and Nine-Species V2 datasets, following previous works. On the other hand, the difference in methodologies successfully leads to the variety in data preference of base models (Fig. 1). These settings meet the criteria for base model selection. The architectures of these six models in detail can be found in the Appendix.

### B.4. ByNovo Details

ByNovo introduces an auxiliary task for identifying key ion peaks alongside the primary peptide sequencing task. Real-world spectral data often contain numerous noise PEAKS and ion PEAKS that are weakly related or unrelated to de novo sequencing, which can interfere with the model's performance (Breci et al., 2003; Tabb et al., 2003). To mitigate this issue, ByNovo incorporates ion peak identification as an auxiliary objective, modeling ion recognition as a token-level classification problem by introducing an ion annotation head within the encoder.

Specifically, ByNovo first labels the ion PEAKS in the spectrum, assigning an ion type $l_i \in \{b/y, \text{other}\}$ to each peak. The task is formalized as maximizing the conditional probability of predicting the ion type $l_i$, given the mass-to-charge ratio $m_i$ and charge $z_i$, as shown in the equation:

$$\mathbf{P}(l_i \mid m_i, z_i, \theta) = \frac{e^{f(m_i, z_i, l_i; \theta)}}{\sum_{l' \in L} e^{f(m_i, z_i, l'; \theta)}} \tag{11}$$

where $f(\cdot)$ is the classification model, and $\theta$ denotes the model parameters. For the entire ion peak sequence in a spectrum, ByNovo maximizes the joint probability, as defined in the equation:

$$\mathbf{P}(\mathbf{L} \mid \mathbf{S}) = \prod_{i=1}^{N} \mathbf{P}(l_i \mid m_i, z_i, \theta) \tag{12}$$

In this classification task, ByNovo uses Focal Loss as the supervision loss function, defined in the equation:

$$\boldsymbol{\mathcal{L}}_{\text{ion}}(p_t) = -(1 - p_t)^{\gamma} \log(p_t) \tag{13}$$

where $p_t$ is the predicted probability of the correct class, and $\gamma > 0$ is a focusing parameter. Focal Loss assigns smaller penalties to well-classified examples with high confidence, while increasing the loss for hard-to-classify samples. This encourages the model to focus on learning difficult examples, which is beneficial for detecting easily overlooked b/y ion PEAKS in spectra.

By explicitly supervising ion peak classification as a token classification task, the model is guided to learn the critical features distinguishing b/y ions from other ions. In complex spectral scenarios, this supervision signal implicitly constrains and regularizes the peptide sequence prediction process, improving sequencing accuracy. This multi-task learning approach helps the model learn more discriminative feature representations, reduces the risk of overfitting, and enhances generalization performance.

## C. Reranking Framework Comparison

| | Objective | Col-Attn | Bacillus | C. bacteria | Honeybee | Human | M.mazei | Mouse | Rice bean | Tomato | Yeast | Average |
|---|---|---|---|---|---|---|---|---|---|---|---|---|
| | Point | ✗ | 0.869 | 0.741 | 0.803 | 0.796 | 0.835 | 0.824 | 0.85 | 0.841 | 0.811 | 0.819 |
| | Pair | ✗ | 0.87 | 0.741 | 0.803 | 0.799 | 0.836 | 0.826 | 0.854 | 0.841 | 0.811 | 0.82 |
| | List | ✗ | 0.865 | 0.737 | 0.799 | 0.786 | 0.829 | 0.825 | 0.832 | 0.844 | 0.817 | 0.815 |
| Amino Acid Precision | PMD+RMD | ✗ | 0.871 | 0.745 | 0.809 | 0.801 | 0.839 | 0.828 | 0.854 | 0.844 | 0.812 | 0.822 |
| | Point | ✔ | 0.871 | 0.74 | 0.806 | 0.8 | 0.837 | 0.825 | 0.852 | 0.841 | 0.812 | 0.82 |
| | Pair | ✔ | 0.871 | 0.742 | 0.804 | 0.798 | 0.837 | 0.825 | 0.853 | 0.842 | 0.812 | 0.82 |
| | List | ✔ | 0.866 | 0.738 | 0.797 | 0.780 | 0.832 | 0.826 | 0.846 | 0.845 | 0.821 | 0.817 |
| | PMD+RMD† | ✔ | **0.874** | **0.746** | **0.81** | **0.802** | **0.84** | **0.828** | **0.859** | **0.844** | **0.816** | **0.824** |
| | Point | ✗ | 0.727 | 0.528 | 0.614 | 0.628 | 0.660 | 0.575 | 0.721 | 0.680 | 0.690 | 0.647 |
| | Pair | ✗ | 0.728 | 0.529 | 0.614 | 0.631 | 0.661 | 0.573 | 0.722 | 0.681 | 0.692 | 0.648 |
| | List | ✗ | 0.725 | 0.540 | 0.622 | 0.613 | 0.663 | 0.590 | 0.694 | 0.670 | 0.705 | 0.646 |
| Peptide Recall | PMD+RMD | ✗ | 0.732 | 0.535 | 0.623 | 0.638 | 0.664 | 0.579 | 0.727 | 0.685 | 0.695 | 0.653 |
| | Point | ✔ | 0.731 | 0.529 | 0.619 | 0.634 | 0.663 | 0.577 | 0.722 | 0.681 | 0.693 | 0.65 |
| | Pair | ✔ | 0.729 | 0.531 | 0.616 | 0.634 | 0.662 | 0.575 | 0.721 | 0.684 | 0.696 | 0.65 |
| | List | ✔ | 0.728 | 0.536 | 0.613 | 0.605 | 0.663 | 0.588 | 0.710 | 0.694 | 0.706 | 0.649 |
| | PMD+RMD† | ✔ | **0.738** | **0.539** | **0.63** | **0.642** | **0.672** | **0.583** | **0.733** | **0.691** | **0.703** | **0.660** |

*Table 5.* Performance comparison on 9-species-V1 test set when the reranking framework varies. The symbol "†" indicates that the model is the final RankNovo mentioned in the main text.

Our RankNovo reranking framework features using the accurate peptide mass deviation metric PMD and RMD as similarity labels. Additionally, we use axial attention (particularly column-wise attention compared to ordinary language models) to boost peptide feature mixing. Here we compare this framework to some established reranking settings in order to prove that RankNovo captures the key modality feature of peptide and mass spectrums and is a superior methodology than those used in NLP tasks on the peptide sequencing task.

Our comparison involves two aspects: the reranking loss level and the backbone model level. We mainly compare RankNovo with the classic RankT5 (Zhuang et al., 2023) framework. RankT5 summarized the three common styles of reranking losses: point-wise, pair-wise, and list-wise loss. Suppose a given query $q_i$ has $N$ potentially relevant candidate documents $d_{i1}, d_{i2}, \ldots, d_{iN}$ to rerank, a reranking framework uses a backbone language model $\mathcal{M}$ to extract the latent $z_{ij}$ representing the relationship between $q_i$ and $d_{ij}$. Then, $z_{ij}$ is projected (in RankT5, the 'projection' is accomplished by learning a new word) to predict the similarity score $\hat{y}_{ij}$. The process can be summarized as:

$$\hat{y}_{ij} = \text{Projection}(\mathcal{M}(q_i, d_{ij})) \tag{14}$$

For the peptide sequencing task, we set the true relevance label $y_{ij}$ as a binary classification label (since our metric PMD and RMD are not adopted). Then the point-wise loss function for each sample $q_i$ equals a summation of binary cross entropy (BCE) losses between each query-document pair.

$$\mathcal{L}_{\text{Point}}(y_i, \hat{y}_i) = - \sum_{j|y_{ij}=1} log(\sigma(\hat{y}_{ij})) - \sum_{j|y_{ij}=0} log(\sigma(1 - \hat{y}_{ij})), \sigma(x) = \frac{1}{1 + e^{-x}} \tag{15}$$

Pairwise reranking loss focuses on enlarging the predicted similarity deviation between relevant query-document pairs and the irrelevant ones, which can be represented as:

$$\mathcal{L}_{\text{Pair}}(y_i, \hat{y}_i) = \sum_{j=1}^{N} \sum_{j'=1}^{N} \mathbb{I}_{y_{ij} > y_{ij'}} log(1 + e^{\hat{y}_{ij'} - \hat{y}_{ij}}) \tag{16}$$

List-wise loss views reranking as a $N$-class classification. And the loss function can be represented as:

$$\mathcal{L}_{\text{List}}(y_i, \hat{y}_i) = - \sum_{j=1}^{N} y_{ij} log(\frac{e^{\hat{y}_{ij}}}{\sum_{j'} e^{\hat{y}_{ij'}}}) \tag{17}$$

For the backbone model $\mathcal{M}$, the encoder-decoder framework is inarguable. The only concern is whether the candidates $d_{ij}$ should be able to 'see' each other. Some pair-wise or list-wise reranking work uses paired candidates' input and a post-ranking procedure. Here, we use column-wise attention modules to enable list-level perception fields because the existing methods for enabling communications between candidates are too diverse, making exhaustive comparison unrealistic.

Detailed results can be found in Table 5. Whether using column-wise attention or not, the best model among point-wise, pair-wise, and list-wise frameworks falls behind at least 0.1 than RankNovo in terms of peptide recall, which achieves an average of 0.660 peptide recall across the nine species. Therefore, RankNovo is more suitable for the sequencing task than common NLP reranking frameworks. It's worth noticing that amino acid recall and peptide precision do not necessarily follow the same trend, especially when the peptide recalls between two models are close, because different models may solve tasks of varying lengths. However, in the peptide sequencing task, the prime concern is whether a spectrum can be identified. Therefore, our analysis focuses on peptide recall, as in Appendix D.

## D. Ablation Study

### D.1. Full benchmark result

In our implementation, RankNovo utilizes six base models: Casanovo-V2, ContraNovo, ByNovo, R-Casanovo, R-ContraNovo, and R-ByNovo. To investigate the relative superiority of these models, we provide the detailed benchmark performance on 9-species-V1 (Table 6) and 9-species-V2 (Table 7). These results are used as a criterion for model selection in the following sections.

### D.2. Base Model Contribution Ablation

In this section, we would like to examine the necessity of each base model to achieve optimal performance, both during training and inference.

**Abalation analysis on selected model subsets** The six base models of RankNovo have dozens of subsets. Therefore, it's impossible to study every combination. Here, in order to study the influence of the number of base models and each model, we select five subsets. As indicated in Table 6, the performance of these six models on 9-species-V1 from poor to strong is: Casanovo-V2, R-Casanovo, R-ByNovo, R-ContraNovo, ContraNovo, and ByNovo. The five subsets are formed by sequentially removing the strongest model until reaching a minimum model number of 2. The details of these five combinations are listed in Table 8.

| | Methods | Bacillus | C. bacteria | Honeybee | Human | M.mazei | Mouse | Rice bean | Tomato | Yeast | Average |
|---|---|---|---|---|---|---|---|---|---|---|---|
| | | *Amino Acid Precision* | | | | | | | | | |
| **Baselines** | PEAKS | 0.719 | 0.586 | 0.633 | 0.639 | 0.673 | 0.600 | 0.644 | 0.728 | 0.748 | 0.663 |
| | DeepNovo | 0.742 | 0.602 | 0.630 | 0.610 | 0.694 | 0.623 | 0.679 | 0.731 | 0.750 | 0.673 |
| | PointNovo | 0.768 | 0.589 | 0.644 | 0.606 | 0.712 | 0.626 | 0.730 | 0.733 | 0.779 | 0.687 |
| | Casanovo | 0.749 | 0.603 | 0.629 | 0.586 | 0.679 | 0.689 | 0.668 | 0.721 | 0.684 | 0.667 |
| **Base Models** | Casanovo V2$^\dagger$ | 0.806 | 0.685 | 0.727 | 0.69 | 0.774 | 0.768 | 0.769 | 0.799 | 0.762 | 0.753 |
| | ContraNovo$^\dagger$ | 0.828 | 0.706 | 0.761 | 0.771 | 0.798 | 0.799 | 0.804 | 0.808 | 0.782 | 0.784 |
| | ByNovo$^\star$ | 0.858 | 0.723 | 0.791 | 0.767 | 0.823 | 0.803 | 0.836 | 0.828 | 0.804 | 0.804 |
| | R-Casanovo$^\star$ | 0.804 | 0.699 | 0.728 | 0.719 | 0.769 | 0.776 | 0.782 | 0.795 | 0.762 | 0.759 |
| | R-ContraNovo$^\star$ | 0.839 | 0.716 | 0.775 | 0.782 | 0.806 | 0.811 | 0.816 | 0.822 | 0.798 | 0.796 |
| | R-ByNovo$^\star$ | 0.855 | 0.724 | 0.794 | 0.762 | 0.821 | 0.81 | 0.835 | 0.831 | 0.762 | 0.799 |
| **Ours** | **RankNovo** | **0.874** | **0.746** | **0.81** | **0.802** | **0.84** | **0.828** | **0.859** | **0.844** | **0.816** | **0.824** |
| | | *Peptide Recall* | | | | | | | | | |
| **Baselines** | PEAKS | 0.387 | 0.203 | 0.287 | 0.277 | 0.356 | 0.197 | 0.362 | 0.403 | 0.428 | 0.322 |
| | DeepNovo | 0.449 | 0.253 | 0.330 | 0.293 | 0.422 | 0.286 | 0.436 | 0.454 | 0.462 | 0.376 |
| | PointNovo | 0.518 | 0.298 | 0.396 | 0.351 | 0.478 | 0.355 | 0.511 | 0.513 | 0.534 | 0.439 |
| | Casanovo | 0.537 | 0.330 | 0.406 | 0.341 | 0.478 | 0.426 | 0.506 | 0.521 | 0.490 | 0.448 |
| **Base Models** | Casanovo V2$^\dagger$ | 0.646 | 0.46 | 0.527 | 0.492 | 0.592 | 0.493 | 0.628 | 0.637 | 0.629 | 0.567 |
| | ContraNovo$^\dagger$ | 0.684 | 0.487 | 0.576 | 0.624 | 0.628 | 0.563 | 0.676 | 0.655 | 0.669 | 0.618 |
| | ByNovo$^\star$ | 0.708 | 0.499 | 0.597 | 0.584 | 0.639 | 0.545 | 0.696 | 0.667 | 0.676 | 0.623 |
| | R-Casanovo$^\star$ | 0.628 | 0.467 | 0.515 | 0.511 | 0.57 | 0.505 | 0.611 | 0.611 | 0.601 | 0.558 |
| | R-ContraNovo$^\star$ | 0.682 | 0.499 | 0.583 | 0.606 | 0.621 | 0.566 | 0.673 | 0.654 | 0.664 | 0.616 |
| | R-ByNovo$^\star$ | 0.703 | 0.493 | 0.59 | 0.554 | 0.637 | 0.543 | 0.685 | 0.659 | 0.629 | 0.610 |
| **Ours** | **RankNovo** | **0.738** | **0.539** | **0.63** | **0.642** | **0.672** | **0.583** | **0.733** | **0.691** | **0.703** | **0.660** |

*Table 6.* Evaluation of RankNovo in comparison to baseline and base methods on the 9-species-V1 test set. Bolded entries indicate the best-performing models. The symbol "$\dagger$" indicates that the model serves as both a baseline and a base model. "$\star$" signifies that the base models were developed and trained by us.

| | Methods | Bacillus | C. bacteria | Honeybee | Human | M.mazei | Mouse | Rice bean | Tomato | Yeast | Average |
|---|---|---|---|---|---|---|---|---|---|---|---|
| **Amino Acid Precision** | Casanovo V2$^\dagger$ | 0.888 | 0.791 | 0.823 | 0.872 | 0.877 | 0.813 | 0.891 | 0.891 | 0.915 | 0.862 |
| | ContraNovo$^\dagger$ | 0.901 | 0.807 | 0.848 | 0.920 | 0.896 | 0.839 | 0.913 | 0.898 | 0.919 | 0.882 |
| | ByNovo$^\star$ | 0.92 | 0.823 | 0.876 | 0.917 | 0.914 | 0.841 | 0.932 | 0.912 | 0.934 | 0.897 |
| | R-Casanovo$^\star$ | 0.876 | 0.804 | 0.814 | 0.891 | 0.867 | 0.821 | 0.881 | 0.891 | 0.898 | 0.860 |
| | R-ContraNovo$^\star$ | 0.909 | 0.815 | 0.865 | 0.923 | 0.901 | 0.849 | 0.919 | 0.907 | 0.925 | 0.890 |
| | R-ByNovo$^\star$ | 0.919 | 0.822 | 0.879 | 0.912 | 0.912 | 0.843 | 0.932 | 0.913 | 0.936 | 0.897 |
| | **RankNovo** | **0.926** | **0.838** | **0.885** | **0.929** | **0.920** | **0.860** | **0.938** | **0.918** | **0.938** | **0.906** |
| **Peptide Recall** | Casanovo V2$^\dagger$ | 0.793 | 0.558 | 0.669 | 0.712 | 0.754 | 0.555 | 0.772 | 0.783 | 0.837 | 0.714 |
| | ContraNovo$^\dagger$ | 0.815 | 0.575 | 0.711 | 0.820 | 0.780 | 0.616 | 0.799 | 0.794 | 0.854 | 0.752 |
| | ByNovo$^\star$ | 0.833 | 0.582 | 0.731 | 0.789 | 0.799 | 0.596 | 0.814 | 0.807 | 0.871 | 0.758 |
| | R-Casanovo$^\star$ | 0.759 | 0.558 | 0.643 | 0.732 | 0.723 | 0.558 | 0.721 | 0.768 | 0.799 | 0.696 |
| | R-ContraNovo$^\star$ | 0.821 | 0.581 | 0.719 | 0.815 | 0.779 | 0.620 | 0.804 | 0.798 | 0.861 | 0.755 |
| | R-ByNovo$^\star$ | 0.831 | 0.585 | 0.729 | 0.781 | 0.794 | 0.589 | 0.815 | 0.803 | 0.873 | 0.756 |
| | **RankNovo** | **0.851** | **0.620** | **0.752** | **0.820** | **0.813** | **0.629** | **0.836** | **0.822** | **0.885** | **0.781** |

*Table 7.* Evaluation of RankNovo in comparison to baseline and base methods on the 9-species-V2 test set. Bolded entries indicate the best-performing models. The symbol "$\dagger$" indicates that the model serves as both a baseline and a base model. "$\star$" signifies that the base models were developed and trained by us.

| | N. Train | N. Infer | Bacillus | C. bacteria | Honeybee | Human | M.mazei | Mouse | Rice bean | Tomato | Yeast | Average |
|---|---|---|---|---|---|---|---|---|---|---|---|---|
| **Amino Acid Precision** | 2 | 2 | 0.832 | 0.721 | 0.752 | 0.735 | 0.795 | 0.785 | 0.804 | 0.806 | 0.796 | 0.781 |
| | 3 | 3 | 0.864 | 0.749 | 0.8 | 0.774 | 0.828 | 0.811 | 0.846 | 0.829 | 0.796 | 0.811 |
| | 4 | 4 | 0.865 | 0.751 | 0.802 | 0.794 | 0.834 | 0.819 | 0.847 | 0.831 | 0.817 | 0.818 |
| | 5 | 5 | 0.87 | 0.756 | 0.804 | 0.802 | 0.835 | 0.825 | 0.852 | 0.832 | 0.822 | 0.822 |
| | 6$^\dagger$ | 6 | **0.874** | **0.746** | **0.81** | **0.802** | **0.84** | **0.828** | **0.859** | **0.844** | **0.816** | **0.824** |
| **Peptide Recall** | 2 | 2 | 0.667 | 0.482 | 0.548 | 0.533 | 0.598 | 0.519 | 0.649 | 0.641 | 0.636 | 0.586 |
| | 3 | 3 | 0.711 | 0.516 | 0.599 | 0.575 | 0.645 | 0.55 | 0.701 | 0.672 | 0.641 | 0.623 |
| | 4 | 4 | 0.72 | 0.525 | 0.613 | 0.613 | 0.656 | 0.568 | 0.711 | 0.679 | 0.673 | 0.64 |
| | 5 | 5 | 0.73 | 0.534 | 0.618 | 0.633 | 0.665 | 0.583 | 0.724 | 0.683 | 0.688 | 0.651 |
| | 6$^\dagger$ | 6 | **0.738** | **0.539** | **0.63** | **0.642** | **0.672** | **0.583** | **0.733** | **0.691** | **0.703** | **0.660** |

*Table 9.* Peptide recall evaluation of RankNovo on 9-species-V1 test set when the training base model set and the inference base model set are the same and vary. The symbol "$\dagger$" indicates that the model is the final RankNovo mentioned in the main text.

Firstly, we consider the impact of some base models being completely disregarded during training and inference. From

| Model Num. | Base Model Set |
|---|---|
| 2 | Casanovo-V2, R-Casanovo |
| 3 | Casanovo-V2, R-Casanovo, R-ByNovo |
| 4 | Casanovo-V2, R-Casanovo, R-ByNovo, R-ContraNovo |
| 5 | Casanovo-V2, R-Casanovo, R-ByNovo, R-ContraNovo, ContraNovo |
| 6 | Casanovo-V2, R-Casanovo, R-ByNovo, R-ContraNovo, ContraNovo, ByNovo |

*Table 8.* Description of different combinations of base models

Table 9, we can see that can more base models that are used, the peptide recall on the 9-species-V1 dataset ascends, from the lowest 0.586 of two models to the highest 0.660 of six models. On the other hand, even when all six models are used during inference, the absence of models during training affects the final performance. As in Table 10, the combination of two base models achieves the lowest peptide recall of 0.649, 1.6% worse than the combination of all models.

| | N. Train | N. Infer | *Bacillus* | *C. bacteria* | *Honeybee* | *Human* | *M.mazei* | *Mouse* | *Rice bean* | *Tomato* | *Yeast* | **Average** |
|---|---|---|---|---|---|---|---|---|---|---|---|---|
| **Amino Acid Precision** | 2 | 6 | 0.870 | 0.754 | 0.806 | 0.803 | 0.837 | 0.821 | 0.856 | 0.833 | 0.826 | 0.823 |
| | 3 | 6 | 0.872 | 0.757 | 0.805 | 0.8 | 0.835 | 0.822 | 0.852 | 0.834 | 0.826 | 0.823 |
| | 4 | 6 | 0.872 | 0.756 | 0.807 | 0.805 | 0.839 | 0.822 | 0.855 | 0.835 | 0.827 | 0.824 |
| | 5 | 6 | 0.875 | 0.761 | 0.81 | 0.806 | 0.839 | 0.826 | 0.858 | 0.837 | 0.828 | 0.827 |
| | 6† | 6 | **0.874** | **0.746** | **0.81** | **0.802** | **0.84** | **0.828** | **0.859** | **0.844** | **0.816** | **0.824** |
| **Peptide Recall** | 2 | 6 | 0.727 | 0.525 | 0.613 | 0.627 | 0.663 | 0.577 | 0.723 | 0.684 | 0.686 | 0.647 |
| | 3 | 6 | 0.731 | 0.533 | 0.617 | 0.628 | 0.661 | 0.579 | 0.719 | 0.685 | 0.693 | 0.649 |
| | 4 | 6 | 0.733 | 0.533 | 0.625 | 0.638 | 0.668 | 0.580 | 0.729 | 0.688 | 0.700 | 0.655 |
| | 5 | 6 | 0.734 | 0.537 | 0.624 | 0.638 | 0.67 | 0.584 | 0.729 | 0.689 | 0.696 | 0.657 |
| | 6† | 6 | **0.738** | **0.539** | **0.63** | **0.642** | **0.672** | **0.583** | **0.733** | **0.691** | **0.703** | **0.660** |

*Table 10.* Peptide recall evaluation of RankNovo on 9-species-V1 test set when the training base model set varies and the inference base model set is fixed. The symbol "†" indicates that the model is the final RankNovo mentioned in the main text.

Combining these two experiments, two important results are shown. Firstly, the impact of not using all models exists, both at training and at the inference stage. This means the integration of more diversity during training enriches the knowledge of RankNovo. Secondly, we can see that the number of models during inference is more important than that during training. Both training with two models, the peptide recall rises by 10.7% when the number of inference models increases from 2 to 6. This shows RankNovo's zero-shot generalization ability, which is more delicately shown in Section E.1.

**Additional results on reversed model subsets** The model subsets combination in Table 8 is further used in Sec 4.4. To verify that sequentially removing the best base model is not the key factor of our findings, we generate a reverse combination of model subsets by sequentially removing the worst model. The generated model subsets are listed in Table 11.

The same experiments as above are conducted on this combination. As shown in Table 12 and Table 13. The overall trend remains consistent under this alternative setting. Knowledge acquired by certain base models can still be adapted to other base models in a zero-shot manner during inference. Increasing the number of base models during training continues to yield better overall performance. However, as expected, the performance trends are less pronounced compared to our original configuration, where the strongest models were sequentially removed. This additional analysis strengthens our understanding of the interplay between base model selection and RankNovo's overall performance.

Since the original settings yield more pronounced performance differences between configurations, it is adapted for further experiments and analysis.

| Model Num. | Base Model Set |
|---|---|
| 2 | ByNovo, ContraNovo |
| 3 | ByNovo, ContraNovo, R-ContraNovo |
| 4 | ByNovo, ContraNovo, R-ContraNovo, R-ByNovo |
| 5 | ByNovo, ContraNovo, R-ContraNovo, R-ByNovo, R-Casanovo |
| 6 | ByNovo, ContraNovo, R-ContraNovo, R-ByNovo, R-Casanovo, Casanovo-V2 |

*Table 11.* Description of different combinations of base models. The combinations are generated by sequentially removing the weakest model.

| | N. Train | N. Infer | *Bacillus* | *C. bacteria* | *Honeybee* | *Human* | *M.mazei* | *Mouse* | *Rice bean* | *Tomato* | *Yeast* | **Average** |
|---|---|---|---|---|---|---|---|---|---|---|---|---|
| **Amino Acid Precision** | 2 | 2 | 0.855 | 0.721 | 0.787 | 0.781 | 0.819 | 0.809 | 0.832 | 0.825 | 0.799 | 0.803 |
| | 3 | 3 | 0.864 | 0.733 | 0.799 | 0.798 | 0.832 | 0.824 | 0.844 | 0.836 | 0.812 | 0.816 |
| | 4 | 4 | 0.868 | 0.737 | 0.806 | 0.8 | 0.836 | 0.825 | 0.85 | 0.84 | 0.812 | 0.819 |
| | 5 | 5 | 0.871 | 0.743 | 0.808 | 0.804 | 0.838 | 0.829 | 0.855 | 0.842 | 0.818 | 0.823 |
| | 6† | 6 | **0.874** | **0.746** | **0.81** | **0.802** | **0.84** | **0.828** | **0.859** | **0.844** | **0.816** | **0.824** |
| **Peptide Recall** | 2 | 2 | 0.707 | 0.501 | 0.599 | 0.617 | 0.644 | 0.563 | 0.703 | 0.667 | 0.684 | 0.632 |
| | 3 | 3 | 0.719 | 0.518 | 0.613 | 0.638 | 0.653 | 0.577 | 0.709 | 0.677 | 0.694 | 0.644 |
| | 4 | 4 | 0.727 | 0.522 | 0.619 | 0.634 | 0.658 | 0.576 | 0.715 | 0.681 | 0.696 | 0.648 |
| | 5 | 5 | 0.732 | 0.530 | 0.624 | 0.638 | 0.663 | 0.582 | 0.727 | 0.685 | 0.702 | 0.654 |
| | 6† | 6 | **0.738** | **0.539** | **0.63** | **0.642** | **0.672** | **0.583** | **0.733** | **0.691** | **0.703** | **0.660** |

*Table 12.* Peptide recall evaluation of RankNovo on 9-species-V1 test set when the training base model set and the inference base model set are the same and vary. The symbol "†" indicates that the model is the final RankNovo mentioned in the main text. The model subsets here are created by sequentially removing the weakest model, as introduced in Table 11.

| | N. Train | N. Infer | *Bacillus* | *C. bacteria* | *Honeybee* | *Human* | *M.mazei* | *Mouse* | *Rice bean* | *Tomato* | *Yeast* | **Average** |
|---|---|---|---|---|---|---|---|---|---|---|---|---|
| **Amino Acid Precision** | 2 | 6 | 0.871 | 0.745 | 0.808 | 0.801 | 0.839 | 0.828 | 0.856 | 0.843 | 0.814 | 0.823 |
| | 3 | 6 | 0.871 | 0.745 | 0.809 | 0.802 | 0.839 | 0.829 | 0.856 | 0.844 | 0.815 | 0.823 |
| | 4 | 6 | 0.873 | 0.745 | 0.810 | 0.802 | 0.841 | 0.829 | 0.857 | 0.845 | 0.817 | 0.824 |
| | 5 | 6 | 0.872 | 0.743 | 0.81 | 0.802 | 0.839 | 0.829 | 0.858 | 0.843 | 0.817 | 0.824 |
| | 6† | 6 | **0.874** | **0.746** | **0.81** | **0.802** | **0.84** | **0.828** | **0.859** | **0.844** | **0.816** | **0.824** |
| **Peptide Recall** | 2 | 6 | 0.727 | 0.526 | 0.615 | 0.627 | 0.665 | 0.581 | 0.726 | 0.685 | 0.687 | 0.649 |
| | 3 | 6 | 0.732 | 0.529 | 0.623 | 0.632 | 0.665 | 0.583 | 0.727 | 0.685 | 0.698 | 0.653 |
| | 4 | 6 | 0.735 | 0.535 | 0.625 | 0.639 | 0.669 | 0.583 | 0.728 | 0.688 | 0.700 | 0.656 |
| | 5 | 6 | 0.738 | 0.533 | 0.628 | 0.639 | 0.673 | 0.584 | 0.735 | 0.690 | 0.701 | 0.658 |
| | 6† | 6 | **0.738** | **0.539** | **0.63** | **0.642** | **0.672** | **0.583** | **0.733** | **0.691** | **0.703** | **0.660** |

*Table 13.* Peptide recall evaluation of RankNovo on 9-species-V1 test set when the training base model set varies and the inference base model set is fixed. The symbol "†" indicates that the model is the final RankNovo mentioned in the main text. The model subsets here are created by sequentially removing the weakest model, as introduced in Table 11.

### D.3. Training Objective Ablation

| | Objective | *Bacillus* | *C. bacteria* | *Honeybee* | *Human* | *M.mazei* | *Mouse* | *Rice bean* | *Tomato* | *Yeast* | **Average** |
|---|---|---|---|---|---|---|---|---|---|---|---|
| **Amino Acid Precision** | RMD | 0.869 | 0.755 | 0.807 | 0.802 | 0.838 | 0.822 | 0.853 | 0.834 | 0.827 | 0.821 |
| | PMD | 0.871 | 0.742 | 0.810 | 0.806 | 0.836 | 0.823 | 0.856 | 0.835 | 0.821 | 0.822 |
| | PMD + RMD† | **0.874** | **0.746** | **0.81** | **0.802** | **0.84** | **0.828** | **0.859** | **0.844** | **0.816** | **0.824** |
| **Peptide Recall** | RMD | 0.731 | 0.529 | 0.618 | 0.632 | 0.664 | 0.576 | 0.723 | 0.684 | 0.691 | 0.65 |
| | PMD | 0.731 | 0.534 | 0.623 | 0.637 | 0.664 | 0.577 | 0.726 | 0.685 | 0.694 | 0.652 |
| | PMD + RMD† | **0.738** | **0.539** | **0.63** | **0.642** | **0.672** | **0.583** | **0.733** | **0.691** | **0.703** | **0.660** |

*Table 14.* Evaluation of performance on 9-species-V1 test set when training under different objectives. The symbol "†" indicates that the model is the final RankNovo mentioned in the main text.

In this work, we introduces two novel metrics, PMD and RMD, as the learning objective of reranking models. Here we conduct the ablation study of the effect of the combined use of these two metrics. As shown in Table 14, using RMD alone achieves the lowest peptide recall of 0.650, while only using PMD alone is better, with a peptide recall of 0.657. The best peptide recall of 0.660 is achieved when both PMD and RMD are used.

### D.4. Model Architecture Ablation

| | Col-Attn | *Bacillus* | *C. bacteria* | *Honeybee* | *Human* | *M.mazei* | *Mouse* | *Rice bean* | *Tomato* | *Yeast* | **Average** |
|---|---|---|---|---|---|---|---|---|---|---|---|
| **AA** | ✗ | 0.871 | 0.745 | 0.809 | 0.801 | 0.839 | 0.828 | 0.854 | 0.844 | 0.812 | 0.822 |
| **Precision** | ✔† | **0.874** | **0.746** | **0.81** | **0.802** | **0.84** | **0.828** | **0.859** | **0.844** | **0.816** | **0.824** |
| **Peptide** | ✗ | 0.732 | 0.535 | 0.623 | 0.638 | 0.664 | 0.579 | 0.727 | 0.685 | 0.695 | 0.653 |
| **Recall** | ✔† | **0.738** | **0.539** | **0.63** | **0.642** | **0.672** | **0.583** | **0.733** | **0.691** | **0.703** | **0.660** |

*Table 15.* Performance comparison of RankNovo on 9-species-V1 test set between using column-wise attention in the peptide feature mixer or not. The symbol "†" indicates that the model is the final RankNovo mentioned in the main text.

The effect of whether using column-wise attention has already been mentioned in Section C. In this section, we emphasize its effect when the training objective is chosen to be the combination of PMD and RMD. From Table 15 we can see that when using column-wise attention modules, the average peptide recall across the nine species rises from 0.653 to 0.660. This shows column-wise attention's contribution to the optimal performance of RankNovo.

## E. Additional Results

### E.1. Analysis of Zero-shot Performance

| | N. Train | N. Infer | *Bacillus* | *C. bacteria* | *Honeybee* | *Human* | *M.mazei* | *Mouse* | *Rice bean* | *Tomato* | *Yeast* | **Average** |
|---|---|---|---|---|---|---|---|---|---|---|---|---|
| | 2 | 2 | 0.832 | 0.721 | 0.752 | 0.735 | 0.795 | 0.785 | 0.804 | 0.806 | 0.796 | 0.781 |
| **Amino** | 2 | 3 | 0.86 | 0.732 | 0.799 | 0.773 | 0.826 | 0.813 | 0.845 | 0.835 | 0.778 | 0.807 |
| **Acid** | 2 | 4 | 0.861 | 0.733 | 0.798 | 0.786 | 0.828 | 0.82 | 0.845 | 0.837 | 0.804 | 0.812 |
| **Precision** | 2 | 5 | 0.864 | 0.737 | 0.801 | 0.797 | 0.832 | 0.825 | 0.849 | 0.839 | 0.807 | 0.817 |
| | **2** | **6** | **0.873** | **0.757** | **0.809** | **0.806** | **0.840** | **0.824** | **0.859** | **0.836** | **0.829** | **0.826** |
| | 2 | 2 | 0.667 | 0.482 | 0.548 | 0.533 | 0.598 | 0.519 | 0.649 | 0.641 | 0.636 | 0.586 |
| | 2 | 3 | 0.707 | 0.51 | 0.596 | 0.579 | 0.642 | 0.551 | 0.705 | 0.671 | 0.64 | 0.622 |
| **Peptide** | 2 | 4 | 0.716 | 0.518 | 0.605 | 0.604 | 0.653 | 0.567 | 0.709 | 0.677 | 0.662 | 0.635 |
| **Recall** | 2 | 5 | 0.722 | 0.525 | 0.611 | 0.628 | 0.661 | 0.579 | 0.72 | 0.681 | 0.68 | 0.645 |
| | **2** | **6** | **0.729** | **0.527** | **0.615** | **0.629** | **0.665** | **0.579** | **0.725** | **0.686** | **0.688** | **0.649** |

*Table 16.* Zero-shot performance of a fixed training base model set of two models on unseen models. The numbers are calculated on the 9-species-V1 dataset.

We demonstrate the zero-shot capability of RankNovo by training it exclusively on predictions from the two lowest-performing base models and progressively incorporating predictions from unseen models into the candidate sets for each spectrum during inference. As shown in Table 16, as the number of inference models increases, the average peptide recall improves, rising from 0.586 with 2 models to 0.649 with 6 models.

## E.2. Analysis of Amino Acid Identification with Similar Masses

The experimental results across the nine species, as illustrated in Figure 5, exhibit a consistent improvement in recall for key amino acids (M(O), Q, F, K) when leveraging RankNovo over the baseline methods, Casanovo V2 and ContraNovo. RankNovo consistently outperforms the baselines across all species, particularly in M(O) and F. The recall improvements are most pronounced in species like yeast, ricebean, and honeybee, where RankNovo demonstrates significant performance gains. These results emphasize the strong generalization capabilities of RankNovo across diverse species and its effectiveness in addressing the ambiguities introduced by amino acids with similar masses. The consistent superiority of RankNovo underscores its potential to advance peptide sequencing, especially within complex biological datasets.

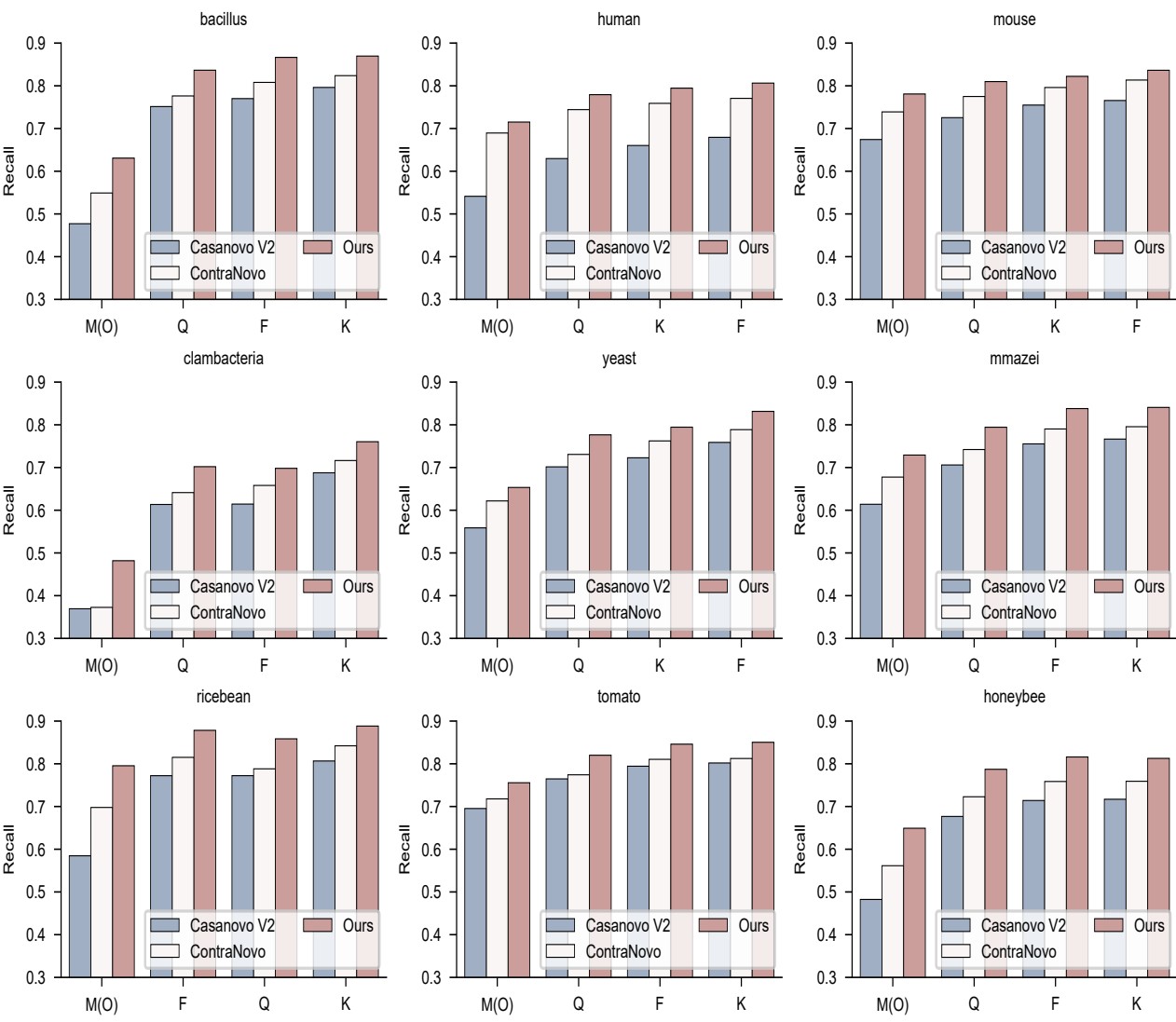

*Figure 5.* The performance comparison of amino acids with similar masses. The numbers are calculated on the 9-species-V1 dataset.

### E.3. Analysis of Peptide Length

The results in Figure 6 demonstrate that our model consistently surpasses the baseline methods, Casanovo V2 and ContraNovo, across a wide variety of species. Specifically, for shorter peptides (lengths 7 to 17), our model achieves significantly higher recall across all species, underscoring its enhanced capacity to capture key sequence patterns in simpler peptide structures. As peptide length increases, performance across all models declines progressively, indicating that longer peptides introduce additional structural complexity that impairs recognition accuracy. Nonetheless, our model maintains a competitive advantage, consistently outperforming the baselines for most species. However, the performance gap diminishes as peptide length increases, likely due to the heightened challenges associated with recognizing longer sequences. These results highlight the effectiveness of our model in processing peptides of varying lengths, as well as its strong generalization capability across diverse species.

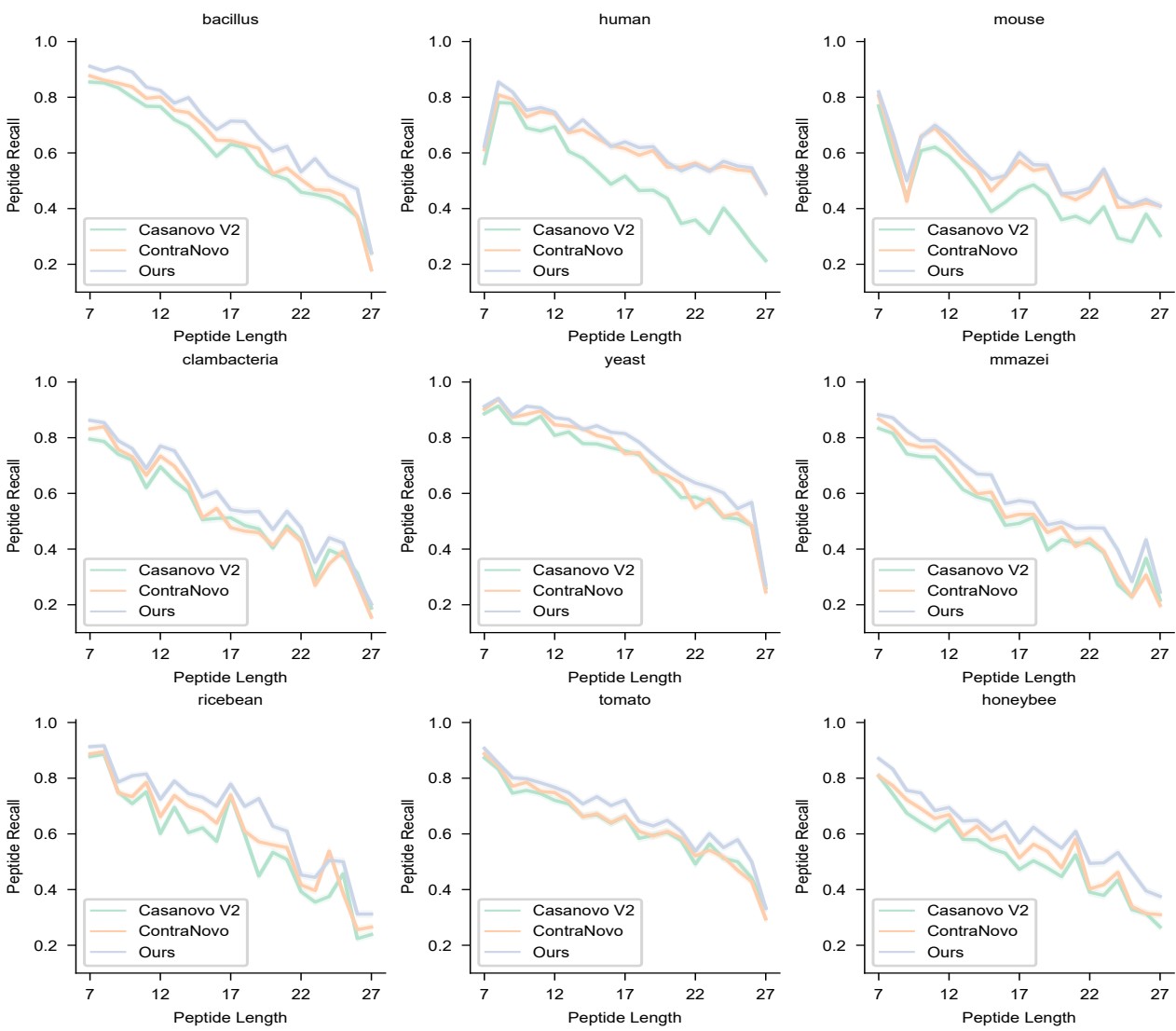

*Figure 6.* Influence of peptide length on 9-species-V1 dataset.

### E.4. Contribution of Each Base Model

By analyzing the contributions of individual base models across nine species, we uncover distinct patterns of efficacy, as depicted in Figure 7. Each base model exhibits varying degrees of influence on RankNovo's peptide selection, underscoring their complementary strengths. Notably, R-ByNovo consistently demonstrates the highest contribution in most species, reaching 41.7% in yeast, while Casanovo-V2 contributes less significantly, particularly in species like tomato and mouse. This variation suggests that different models capture species-specific features with varying effectiveness. The consistent, albeit variable, contributions of each base model highlight the critical importance of model diversity; removing any single model would likely degrade performance for certain species. These findings illustrate the robustness of the ensemble approach, where integrating multiple models compensates for the limitations of individual ones, enabling RankNovo to generalize effectively across a broad range of species and peptide structures.

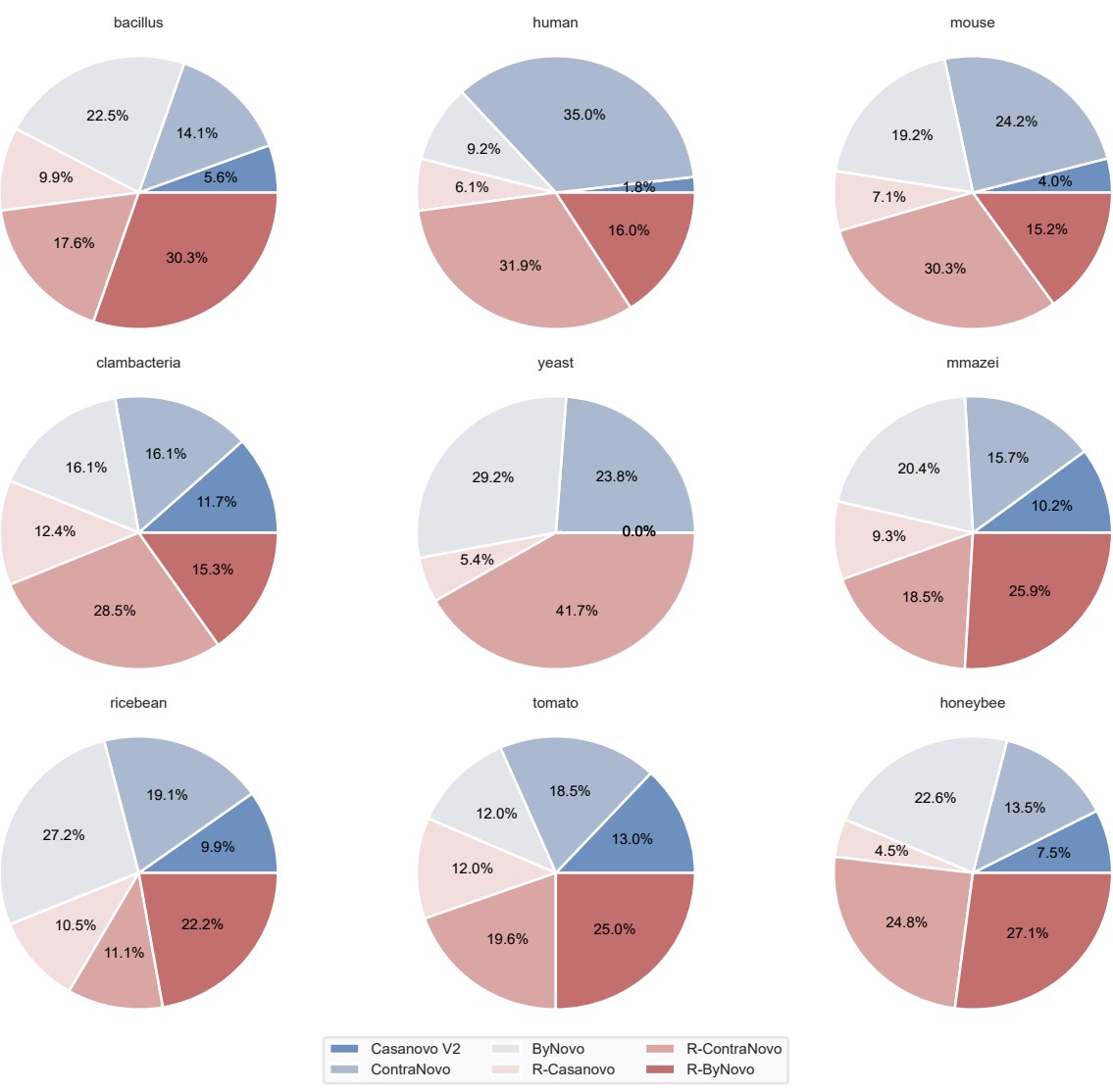

*Figure 7.* Unique-correctly selected percentage of base models. The numbers are calculated on the 9-species-V1 dataset.

## E.5. More information about training time and inference cost

| Model | Parameters (M) | Training Time (Day) | Infer. Cost (s/spectra) | Infer. Speed (spectras/s) |
|---|---|---|---|---|
| Casa. & R-Casa. | 47.3 | 3 | 0.127 | 7.87 |
| Contra. & R-Contra. | 68.6 | 4 | 0.173 | 5.78 |
| By. & R-By. | 49.7 | 3 | 0.169 | 5.92 |
| **RankNovo** | **50.5** | **4** | **/** | **/** |

*Table 17.* Summary of model parameters, training time, and inference speed of six base models and RankNovo.

| N. Infer | Candidates Collection (s/spectra) | Reranking (s/spectra) | Total Cost (s/spectra) | Infer. Speed (spectra/s) |
|---|---|---|---|---|
| 2 | 0.254 | 0.004 | 0.258 | 3.88 |
| 3 | 0.423 | 0.006 | 0.429 | 2.33 |
| 4 | 0.596 | 0.008 | 0.604 | 1.66 |
| 5 | 0.769 | 0.010 | 0.779 | 1.28 |
| 6 | 0.938 | 0.011 | 0.949 | 1.05 |

*Table 18.* RankNovo's inference speed when using different numbers of base models. The combination of base models refers to Table 8.

As shown in Table 17, RankNovo comprises 50.5M parameters and requires 4 days of training utilizing four 40GB A100 GPUs, which is comparable to the base models. Since RankNovo is a reranking framework, its inference speed is anticipated to be slower than single-model approaches. However, the reranking process itself is not the primary time constraint. The majority of RankNovo's inference time is consumed in gathering peptide candidates from base models (Table 17), as these require sequential autoregression and beam search decoding, while RankNovo's inference involves only a single attention forward pass.

As we scale from 2 to 6 base models in RankNovo, the inference time increases approximately linearly, with inference speed decreasing proportionally (Table 18). This increased computational cost is an inherent characteristic of reranking frameworks and represents an unavoidable trade-off compared to single-model approaches. However, RankNovo effectively leverages this additional inference time to achieve superior performance levels unattainable by single models. This inference time-performance trade-off can be flexibly adjusted by modifying the number of candidates.

In the context of denovo peptide sequencing, RankNovo's significance lies in introducing a novel approach that allows researchers to optionally scale up inference time in exchange for enhanced performance. This represents the first such option in the field.

## E.6. Additional Results on PTM identifications

In our current 9-species-V1 benchmark, classes of post-translation modifications (PTMs) are included: Oxidation (M), Deamidation (N), and Deamidation (Q). The promising results of these modifications demonstrated that RankNovo can effectively enhance performance on PTM-containing spectra when such modifications are incorporated during training.

To better verify RankNovo's ability on more diverse PTM types, we conducted additional experiments on a more diverse set of biologically significant PTMs from the dataset compiled by Zolg et al (Zolg et al., 2018). Specifically, we selected three functionally important modifications: Acetylation (K), Dimethylation (K), and Phosphorylation (Y). Each PTM included 62.5K spectra split 8:1:1 for training/validation/testing.

Given that these new PTMs were not in the original vocabulary of our models, we performed necessary fine-tuning procedures. We combined the training and validation datasets across all three PTMs, reinitialized the embedding and final linear layers to accommodate the expanded vocabulary, and fine-tuned both the six base models and RankNovo accordingly.

| PTM | Casanova | ContraNovo | ByNovo | R-Casanova | R-ContraNovo | R-ByNovo | RankNovo |
|---|---|---|---|---|---|---|---|
| Acetylation (K) | 0.819 | 0.820 | 0.830 | 0.833 | 0.806 | 0.832 | 0.889 |
| Dimethylation (K) | 0.455 | 0.459 | 0.458 | 0.458 | 0.401 | 0.457 | 0.487 |
| Phosphorylation (Y) | 0.476 | 0.473 | 0.520 | 0.491 | 0.519 | 0.522 | 0.589 |

*Table 19.* Performance comparison of different models across various post-translational modifications (PTMs).

Our experimental results demonstrate RankNovo's consistent superiority across multiple post-translational modifications (PTMs). As shown in Table 19, RankNovo achieved significant improvements over the best base models: 5.6% for Acetylation (K) (0.889 vs 0.833), 2.8% for Dimethylation (K) (0.487 vs 0.457), and 6.7% for Phosphorylation (Y) (0.589 vs 0.522). These compelling performance gains validate that our deep learning reranking framework maintains its effectiveness across a diverse spectrum of PTMs, highlighting the robustness and broader applicability of our approach for advanced proteomics research.

