# OpenReview forum: "Universal Biological Sequence Reranking for Improved De Novo Peptide Sequencing"
_ICML.cc/2025/Conference — ICML 2025 poster_

### Official Review · Reviewer_EW3W · 2025-03-09

**Overall Recommendation:** 2

**Summary:**

This paper presents RankNovo, a deep reranking framework for de novo peptide sequencing that integrates multiple sequencing models using a listwise reranking approach and axial attention to extract informative features. The introduction of PMD (Peptide Mass Deviation) and RMD (Residual Mass Deviation) provides fine-grained supervision, and extensive experiments show that RankNovo outperforms its base models while achieving state-of-the-art performance with strong zero-shot generalization. The paper is well-structured and presents a compelling methodology, but some areas should be improved.

**Claims And Evidence:**

Yes, The manuscript presents a comprehensive set of experimental results that include comparisons with baseline models, conduct of ablation studies, and exploration of hyperparameter variations.

**Essential References Not Discussed:**

None.

**Experimental Designs Or Analyses:**

Table 1 presents the performance of **RankNovo** and baseline models on the **9-species-v1** and **9-species-v2** test sets.
Figure 3 illustrates the **zero-shot performance** of RankNovo.
Table 3 and Figure 4 show the results of the **ablation study**.

**Methods And Evaluation Criteria:**

Yes, it does.

**Other Comments Or Suggestions:**

None

**Other Strengths And Weaknesses:**

Strengths :
1. The paper is written very clearly and is easy to understand.
2. It achieves SOTA performance on two datasets.
3. The presentation is good.

Weaknesses:
1. Statistical Significance Analysis: While RankNovo outperforms other baselines as shown in Table 1, the paper does not mention whether the improvements are statistically significant. Including a significance analysis (e.g., t-tests or confidence intervals) would strengthen the reliability of the reported results.

2. Lack of Hyperparameter Experiments: Although the appendix provides a hyperparameter table, key parameters such as λ (and potentially others) should be analyzed through dedicated experiments to assess their impact on model performance. A sensitivity study would help determine how robust RankNovo is to different hyperparameter choices.

3. Computational Complexity and Time Efficiency: While RankNovo appears to be a promising approach, the paper lacks a detailed analysis of its computational cost. Given its reliance on multiple base models during training and inference, a discussion on computational complexity and runtime is essential to provide a more comprehensive understanding of its feasibility in real-world applications.

4. Clarification of Zero-shot Experiment Motivation: The zero-shot experimental setup needs clearer justification. It is unclear how this experiment directly relates to the main challenge mentioned in Lines 80–82, which describes the difficulty arising from the heterogeneous nature of mass spectrometry data. Is the zero-shot setting intended to evaluate the model's generalization ability? If so, should it be framed as an OOD (Out-of-Distribution) test instead? A clearer connection between the experimental design and the stated challenge would improve the overall coherence of the study.

**Questions For Authors:**

Please refer to Other Strengths And Weaknesses.

**Relation To Broader Scientific Literature:**

The manuscript presents a clear and well-structured motivation，and and introduces two new metrics, PMD and RMD, to characterize quality differences at the peptide and residue levels.

**Theoretical Claims:**

Not a  theory paper.

---

> ### Author Rebuttal · Authors · 2025-04-01
>
> Thank you for your detailed comments. We address your concerns as follows:
>
> > A significance analysis is needed to strengthen the reliability of performance improvement.
>
> Thank you for your suggestion. We agree that statistical significance analysis is essential for validating the reported improvements. In our main results, we presented only the mean statistical improvements without assessing their significance. ```To address this limitation, we conducted pairwise t-tests at the peptide level to evaluate metric improvements on the 9-species-v1 dataset.``` The t-values for the six base models are 82.4, 42.1, 38.6, 88.2, 43.2, and 50.7, respectively. For all models, the p-values are less than 1e-12. The results show that RankNovo's performance improvements over all base models are **statistically significant**:
> - All comparisons yielded t-statistics of at least 38.6943 (based on total 90K samples)
> - All p-values were below 1e-12, substantially lower than the conventional 0.05 threshold
> These findings conclusively demonstrate that RankNovo achieves statistically significant improvements in both peptide recall and amino acid precision, providing stronger support for the conclusions presented in Section 4.2 (Main Results).
>
> > key parameters such as λ (and potentially others) should be analyzed through dedicated experiments to assess their impact on model performance.
>
> Thank you for your valuable suggestion. While we provided λ evaluations in **Section 4.4** and **Appendix D.3**, we agree more analysis strengthens our work.
> We conducted additional experiments testing different λ values:
> |λ|0|0.25|0.5|0.75|1|
> |----------|-------|-------|-------|-------|-------|
> |pep recall|0.650|0.656|0.660|0.657|0.652|
>
> Our findings show:
> - λ **impacts performance**, with neither extreme values being optimal, validating our choice of λ=0.5 which balances PMD and RMD contributions.
> - Even with sub-optimal λ values, RankNovo maintains peptide recall ≥0.650, still outperforming both ByNovo (0.623) and ContraNovo (0.618), demonstrating the framework's **inherent robustness**.
> ```While computational constraints prevented exhaustive hyperparameter analysis, our experiments across five λ values provide strong evidence for the effectiveness of our selected configuration and the overall robustness of RankNovo's approach.```
>
> > a discussion on computational complexity and runtime is essential to provide a more comprehensive understanding of its feasibility in real-world applications.
>
> Thank you for this important question.
> **Appendix E.5** has already discussed RankNovo's inference speeds. As expected, RankNovo is slower than single-model approaches—a natural trade-off for reranking frameworks. The main bottleneck isn't reranking itself but gathering peptide candidates from base models, which require sequential autoregression and beam search, while RankNovo's inference needs just one attention forward pass.
>
> RankNovo offers a flexible speed-performance trade-off addressing real-world concerns. Users can:
> 1.Use fewer base models/candidates when prioritizing speed
> 2.Scale up when maximum accuracy is critical
> These findings are documented in **Appendix D.2 Table 12**. This adaptability is valuable in proteomics research and clinical settings where resource constraints and accuracy requirements vary significantly, enabling researchers to make informed decisions based on their specific constraints and objectives.
>
> > The zero-shot experiment lacks clear justification. How does it relate to the challenge of heterogeneous mass spectrometry data (Lines 80-82)? Is it testing generalization ability? Should it be framed as an OOD test?
>
> Thank you for this important question about our zero-shot experimental setup. Our zero-shot experiments evaluate RankNovo's generalization to unseen base models, which indeed constitutes an OOD test as you correctly suggested. We train RankNovo using outputs from only a subset of base models, then test its ability to rerank predictions from all models, including previously unseen ones. ```This capability is crucial for practical applications. As new sequencing models emerge, users can apply our released checkpoint directly without retraining, ensuring RankNovo's long-term utility.```
>
> Regarding the heterogeneous MS data challenge, this is primarily addressed by **our reranking approach itself**. We deliberately employed six heterogeneous base models, each capable of performing well on different segments of the heterogeneous data (Fig. 1B). RankNovo functions as a meta-reranker that leverages collective knowledge while minimizing individual biases, directly addressing heterogeneous data distribution challenges, which is the advantage of ensemble learning proven in previous studies [1-3].
>
> [1]End-to-end training of CNN ensembles for person re-identification
> [2]Don't take the easy way out: Ensemble based methods for avoiding known dataset biases
> [3]Exploring model learning heterogeneity for boosting ensemble robustness

---

### Official Review · Reviewer_wxvh · 2025-03-12

**Overall Recommendation:** 4

**Summary:**

The paper outlines a new reranking strategy, wherein the proposed meta-model is capable of ranking candidate peptide sequences for a given tandem spectrum. The proposed meta-model, RankNovo, innovates on existing approaches by (a) using axial attention to derive latent space representations, and (b) utilizing two novel metrics, PMD and RMD as supervision signals. The selected metrics enable a precise quantification of the predicted and true peptide sequences. Results on two benchmark datasets demonstrate the superiority of RankNovo over existing methods for de novo peptide sequencing from mass spectrometry data.

## Update after rebuttal:
I had assigned a score of 4 to the paper after my original review and the other reviews and author rebuttals have not changed my opinion, as I still believe that this work is a valuable contribution to this field.

**Claims And Evidence:**

The paper primarily makes two major claims:

1.	RankNovo surpasses the performance of existing approaches to de novo peptide sequencing. This is substantiated by extensive results in the main paper and the appendices.
2.	RankNovo is able to generalize to unseen models without any additional training. The robustness of RankNovo to base models is is borne out by both the ablation study and the zero-shot performance benchmark.

**Essential References Not Discussed:**

None that I can think of.

**Experimental Designs Or Analyses:**

The experimental design appears to be sound with standard benchmark datasets used. The authors also report residue-level metrics in an effort to be consistent with existing literature. However, my qualm about the use of benchmark models as base models applies here too.

**Methods And Evaluation Criteria:**

RankNovo is benchmarked against 7 other models, some of which were also included as based models, on benchmark datasets. Amino acid precision and peptide recall are reported as metrics when comparing performance. The use of these metrics is justified in Section 4.1, with the peptide recall seemingly the more important metric for the task of de novo sequencing, but amino acid precision being used for completeness’ sake.

I found the evaluation to be fairly comprehensive and pertinent to the task at hand. But I have an issue with the choice of benchmark models. Specifically, the incorporation of base models as a benchmark appears to be “setting the stage” favorably for RankNovo; Casanovo V2, ContraNovo and ByNovo are included among the base models and consistently rank among the best performing benchmarks. Hence, it is not apparent if the superior performance of RankNovo over the other models is more attributable to the base models themselves being better, or the ability of the RankNovo framework to rerank peptides.

**Other Comments Or Suggestions:**

1.	After Eq. 3, when forming the MSA embedding, it is not clear if the stacking is row-wise or column-wise.
2.	The description of the backbone of RankNovo in Section 3.4 would be easier to understand if it were presented in the form of a diagram.

**Other Strengths And Weaknesses:**

I found the ideas presented in this paper interesting and well-grounded. In particular, the use of RMD and PMD to provide supervision signals is logical but innovative and could potentially be utile to other works in the peptide-spectrum space. I also appreciate the thoroughness of the evaluation, with the inclusion of an ablation study and zero-shot performance.

On the other hand, the paper is a little difficult to follow in places. At several points, it was a struggle to make sense of what the authors meant, which detracted from the overall idea and innovations presented therein. I have pointed out specific instances in the next section; many of these comments pertain to dangling references or missing explanations of notations.

**Questions For Authors:**

1.	In the last paragraph of the introduction, the authors state that RankNovo is “the first deep learning-based reranking framework”. I think it is worth reiterating or clarifying that it is the first DL-based reranking framework since some of the base models included already feature deep learning architectures.
2.	Just after Eq. 1, it is stated that “Intensity signals are projected to d dimension with a linear layer because of its relatively lower accuracy”. What does “its” refer to here – the linear layer or intensity signal?
3.	What does $\mathbf{r_i}$ refer to in Eq. 5?

**Relation To Broader Scientific Literature:**

The paper addresses a pertinent need in existing literature around de novo peptide sequencing. Scoring and ranking peptides based on the observed spectrum remains a difficult problem. Hence, a post-hoc reranking approach could be a great addition to the current toolbox for de novo peptide sequencing; rather than just another approach, RankNovo leverages the predictions of existing approaches on its way to delivering a more accurate peptide ranking.

**Theoretical Claims:**

There are no theoretical claims included in this work.

---

> ### Author Rebuttal · Authors · 2025-04-01
>
> Thank you for your detailed comments. We address your concerns as follows:
>
> > Is the choice of base models making benchmark comparison favorably for RankNovo? Should performance improvement be attributed to base models' capabilities or the reranking model?
>
> Thank you for this important question about the contribution of base models versus our reranking approach.
>
> Our benchmark comparison is methodologically sound because: 1) We included all available baselines in our evaluation, **including the previous SOTA ContraNovo**. 2) Our methodology **follows established practices in ensemble research** [1][2][3], where comparing ensemble methods against constituent base models is standard.
>
> To address whether RankNovo's performance stems from stronger base models or the reranking approach itself, our **ablation studies in Appendix D.2 and Table 9** are informative. When training and evaluating RankNovo using only five weaker models, our reranking approach still achieved 0.651 peptide recall - significantly **outperforming the excluded strongest ByNovo model** (0.623). ```This confirms that while high-performing base models contribute to results, the reranking methodology itself provides substantial independent value.```
>
> [1] Routing to the Expert: Efficient Reward-guided Ensemble of Large Language Models
>
> [2] Llm-blender: Ensembling Large Language Models with Pairwise Ranking and Generative Fusion
>
> [3] End-to-end Training of CNN Ensembles for Person Re-identification
>
> > After Eq. 3, when forming the MSA embedding, it is not clear if the stacking is row-wise or column-wise.
>
> Thank you for this important clarification question.
>
> The stacking is performed **row-wise**. For a single peptide sequence embedding with dimensions (L, D)—where L represents the number of amino acids (after padding) and D represents the embedding dimension—the MSA embedding is constructed by row-wise stacking the embeddings of N peptide candidates to form a tensor of shape (N, L, D).
>
> This MSA embedding construction **follows established practices** in the field, consistent with works such as MSA Transformer [1], AlphaFold2 [2], and AlphaFold3 [3].
>
> [1] MSA Transformer
>
> [2] Highly Accurate Protein Structure Prediction with AlphaFold
>
> [3] Accurate Structure Prediction of Biomolecular Interactions with AlphaFold 3
>
> > The description of the backbone of RankNovo in Section 3.4 would be easier to understand if it were presented in the form of a diagram.
>
> Thank you for this constructive suggestion. While we included a model backbone illustration in **Figure 2(B)**, we acknowledge it doesn't fully capture all technical details of RankNovo's architecture.
>
> To address this,  we've created a more detailed diagram focusing on **tensor shapes and attention mechanisms**, available at <https://anonymous.4open.science/r/RankNovo-F2FB/NewFig.png>.
>
> > I think it is worth reiterating or clarifying that RankNovo is the first DL-based reranking framework since some of the base models included already feature deep learning architectures.
>
> Thank you for this important point. You're right - some base models do incorporate deep learning architectures.
>
> But To clarify: ```RankNovo is novel as the first deep learning-based reranking framework specifically for de novo peptide sequencing. While existing models (including the base models) directly decode peptides through regression, RankNovo introduces a fundamentally different approach by evaluating multiple peptide candidates and selecting the optimal prediction.```
>
> This reranking mechanism represents a methodological shift, allowing RankNovo to function as a **universal enhancement module** that integrates with various de novo sequencing models regardless of their architecture.
>
> > Just after Eq. 1, it is stated that “Intensity signals are projected to d dimension with a linear layer because of its relatively lower accuracy”. What does “its” refer to here – the linear layer or intensity signal?
>
> Thank you for highlighting this ambiguity. "Its" refers to the **intensity signal**, not the linear layer.
>
> Intensity signals in MS/MS spectra inherently have lower accuracy compared to m/z signals due to measurement errors, as documented in Chang C, et al. [1]. Given this limitation, we follow Casanovo and ContraNovo by embedding intensity signals using a simple linear transformation, which is less sensitive to tiny changes than the sinusoidal encoding used for m/z signals.
>
> [1] Quantitative and In-Depth Survey of the Isotopic AbundanceDistribution Errors in Shotgun Proteomics
>
> Q6: What does ri refer to in Eq. 5?
>
> A6: In Equation 5:
>  $$g = E_{i \neq j} [P(r_i, r_j)] = \frac{1}{n(n-1)} \sum_{i=1}^{n} \sum_{j=1, j \neq i}^{n} |M(r_i) - M(r_j)|$$
> The term $r_i$ refers to **the i-th residue (amino acid) in our vocabulary**. This equation defines the gap penalty $g$ as the average mass difference between all possible pairs of non-identical amino acids.

---

### Official Review · Reviewer_4F4K · 2025-03-12

**Overall Recommendation:** 3

**Summary:**

This paper presents RankNovo, the deep reranking framework that enhances de novo peptide sequencing by leveraging the complementary strengths of multiple sequencing models.

**Claims And Evidence:**

The claims made in the submission are supported by clear and convincing evidence.

**Essential References Not Discussed:**

- The key contribution of this paper is ranking the output of de novo peptide sequencing models conditioned on the spectral information. However, the final results can only be selected from the candidate peptides, which limits flexibility. SearchNovo [1] leverages spectral information for database searching and uses the retrieved candidate peptides to enhance de novo peptide sequencing, making it more flexible in comparison. This aspect is not discussed in the paper.

[1] Bridging the Gap between Database Search and De Novo Peptide Sequencing with SearchNovo, bioRxiv  2024.

**Experimental Designs Or Analyses:**

- The comparative methods in this paper should include more baseline for a more comprehensive comparison[1][2][3][4].


[1] De novo peptide sequencing with InstaNovo: Accurate, database-free peptide identification for large scale proteomics experiments.

[2] AdaNovo: Adaptive De Novo Peptide Sequencing with Conditional Mutual Information.

[3] π-PrimeNovo: An Accurate and Efficient Non-Autoregressive Deep Learning Model for De Novo Peptide Sequencing.

[4] π-HelixNovo for practical large-scale de novo peptide sequencing.

**Methods And Evaluation Criteria:**

### Dataset
- Although the model has been tested on nine-species v1 & v2, it is better to evaluate it on the latest benchmark datasets [1], which include more diverse data.

### Method
- Table 5 shows that applying col-wise attention results in only a minor improvement (0.007 in Avg. Peptide Recall) compared to models without it while increasing computational complexity.
- Tables 17 and 18 show that RankNovo's inference speed is approximately 6–8 times slower than that of the base model.
- The innovation of the method is limited. It introduces the rerank to the de novo peptide sequencing. However, this task requires accurate inference rather than simply ranking peptides. **If none of the peptide sequences output by the base models are exact matches, the final output of the model will also be incorrect sequences, thus compromising the flexibility of de novo peptide sequencing.**

[1] NovoBench: Benchmarking Deep Learning-based De Novo Peptide Sequencing Methods in Proteomics, NeurIPS 2024.

**Other Comments Or Suggestions:**

### Typo
- In Table 15, the Avg. Peptide Recall values are 0.653. However, in Line 1010, it is written as 0.657.

**Other Strengths And Weaknesses:**

- The paper is written clearly.

**Questions For Authors:**

- Fix typos.
- Evaluate on the latest benchmark datasets.
- Add more baseline models for comparison.
- Explain the advantages of the ranking technique in de novo peptide sequencing compared to the latest database-enhanced method, SearchNovo.

**Relation To Broader Scientific Literature:**

- This paper introduces the rerank technique from NLP to the field of de novo peptide sequencing.

**Theoretical Claims:**

This paper does not have a theoretical part, so there is no need to check it.

---

> ### Author Rebuttal · Authors · 2025-04-01
>
> Thank you for your detailed comments. We address your concerns as follows:
>
> > It's better to provide benchmark performance of RankNovo on NovoBench.
>
> Thank you for your suggestion. Following your recommendation, we expanded our evaluation to include all three **NovoBench** data sources: **Seven-Species (from DeepNovo), Nine-Species-V1, and HC-PT (from InstaNovo).** We construct Seven-Species and HC-PT as instructed in NovoBench.
>
> | Model | Nine-Species (yeast) | Seven-Species (yeast) | HC-PT |
> |-------|---------------------|----------------------|-------|
> | Casanovo | 0.48 | 0.12 | 0.21 |
> | Instanovo | 0.53 | - | 0.57 |
> | AdaNovo | 0.50 | 0.17 | 0.21 |
> | HelixNovo | 0.52 | 0.23 | 0.21 |
> | SearchNovo | 0.55 | 0.26 | 0.45 |
> | ByNovo (best base model) | 0.68 | 0.03 | 0.82 |
> | RankNovo | 0.70 | 0.04 | 0.89 |
>
> The results on NovoBench show that ```RankNovo is consistently superior on Nine-Species and HC-PT datasets.``` On the other hand, zero-shot performance of models trained on MassiveKB (ByNovo and RankNovo) don't perform normally on Seven-Species. **Further analysis shows that the reason lies in the distribution misalignment between MassiveKB and Seven-Species.** The former is collected by high-resolution MS equipment, while the latter is composed of low-resolution data, which makes the phenomenon explainable.
>
> ```We greatly appreciate your valuable suggestion and will incorporate these additional benchmark results and the discussion about data composition into our camera-ready manuscript.```
>
> > More baselines should be included for a more comprehensive comparison.
>
> Thank you for this valuable suggestion. We have expanded our evaluation to include **InstaNovo, AdaNovo, π-PrimeNovo, and π-HelixNovo**. Results show **RankNovo outperforms all these baselines across all metrics on Nine-Species-V1**, a brief summary is provided below.
>
> | Model | Avg. Peptide Recall |
> |-------|---------------------|
> | Casanovo | 0.481 |
> | InstaNovo | 0.532 |
> | AdaNovo | 0.505 |
> | PrimeNovo | 0.638 |
> | HelixNovo | 0.517 |
> | RankNovo | 0.660 |
>
> Additionally, as mentioned in our response to Q1, **we've included comprehensive comparisons with InstaNovo, AdaNovo, and π-HelixNovo on NovoBench.**
>
> **These expanded baselines and comparisons will be included in the camera-ready version.**
>
> > It's recommended to discuss works combining database search and de novo sequencing,  such as SearchNovo.
>
> Thank you for this valuable recommendation. We have expanded **Section 2.1** to include **SearchNovo**, a concurrent development that combines database search with de novo sequencing. SearchNovo employs a retrieval mechanism to identify similar peptide-spectrum matches and integrates prefix peptide sequence information through a fusion layer.
> This addition provides information on the broader research landscape. **We will include this expanded discussion in the camera-ready version.**
>
> > RankNovo requires the target peptide predicted by one of the base models and this may harm flexibility. The advantages of reranking methods compared to SearchNovo should be explained.
>
> Thank you for this important question about flexibility limitations and advantages.
>
> ```We view SearchNovo and RankNovo as complementary approaches with distinct advantages:```
>
> - **Inference-time vs. Training-time Improvement**
> While SearchNovo enhances itself through novel architectures and training techniques, RankNovo explores extracting additional value from existing model outputs at inference time. As models approach optimal performance on available training data, **the collective information in prediction data represents an untapped resource that RankNovo specifically leverages.**
>
> - **Training-free Scaling and Adaptation**
> SearchNovo primarily uses information from the single most similar PSM, limiting its capabilities post-training. In contrast, RankNovo's axial-attention modules integrate features from multiple base models and demonstrate training-free performance scaling with more candidates **(Section 4.3, Fig 3(A), Table 12)**. This allows practitioners to balance efficiency and performance as needed.
>
> - **Controlled Prediction Space with Enhanced Reliability**
> While RankNovo requires at least one base model to correctly sequence the peptide, **this constraint brings practical advantages**. By limiting modifications to existing predictions rather than generating novel sequences, **RankNovo minimizes the risk of introducing new errors**, **providing heightened reliability and interpretability** for real-world applications. Despite this constrained prediction space, **RankNovo achieves state-of-the-art performance across all benchmarks.**
>
> In conclusion, these approaches address different aspects of sequencing challenge and can be integrated - SearchNovo could serve as a base model within RankNovo's framework, potentially combining respective advantages.
>
> > Fix typos.
>
> Sorry for the confusion, the typos will be fixed in the final script.

---

> > ### Comment · Reviewer_4F4K · 2025-04-03
> >
> > Thank you for the reviewer's feedback. I have improved my score.

---

### Official Review · Reviewer_Rt9T · 2025-03-13

**Overall Recommendation:** 3

**Summary:**

This paper introduces RankNovo, an innovative deep reranking framework designed to enhance de novo peptide sequencing accuracy. RankNovo leverages the complementary strengths of multiple sequencing models to overcome the limitations of individual approaches.

RankNovo represents the first deep learning-based reranking framework for de novo peptide sequencing, introducing several innovations: it models candidates as multiple sequence alignments with axial attention to extract cross-candidate features while reducing computational complexity; develops mass-focused metrics (PMD and RMD) that provide precise supervision by quantifying differences at both sequence and residue levels; achieves state-of-the-art performance on benchmark datasets, surpassing its strongest base model by 6.1% and the previous best method by 4.3% in peptide recall; demonstrates robust zero-shot generalization to unseen models; and significantly improves discrimination between amino acids with similar masses, addressing a key challenge in mass spectrometry-based peptide identification.

The dual-track architecture combines spectrum feature extraction via a Transformer encoder with peptide candidate processing through axial attention, joined by a cross-attention mechanism. By jointly optimizing PMD and RMD objectives, RankNovo offers a flexible trade-off between inference time and performance, advancing the frontier of accurate de novo peptide sequencing.

**Claims And Evidence:**

The majority of RankNovo's claims are well-supported by evidence, with comprehensive experiments and analyses. The paper convincingly demonstrates state-of-the-art performance through extensive benchmarking on standard datasets, with Tables 1-2 showing clear improvements over previous methods; the effectiveness of the novel PMD and RMD metrics is verified through ablation studies in Table 3; the complementary contributions of base models are illustrated through detailed analysis in Fig. 3(B) and Fig. 7; and improved discrimination between similar-mass amino acids is supported by comprehensive data in Fig. 3(D) and Fig. 5.

However, regarding zero-shot generalization, while Fig. 3(A) demonstrates RankNovo's ability to rerank predictions from unseen models, the paper lacks validation on more challenging datasets with diverse post-translational modifications (PTMs). Testing on specialized PTM-rich datasets, or synthetic phosphopeptide collections would provide stronger evidence of true generalization capability across the complex PTM landscape that characterizes real-world proteomics applications. This additional validation would strengthen the claim that RankNovo can generalize to difficult cases beyond the standard benchmarks.

**Essential References Not Discussed:**

The RankNovo paper, while generally thorough in its literature review, overlooks several important research directions that would better contextualize its contributions. In proteomics, the paper doesn't acknowledge ensemble approaches like "PepExplorer" (Leprevost et al., 2014) and "iProphet" (Shteynberg et al., 2011), which pioneered the combination of multiple search engines for peptide identification. Similarly, when discussing PTM analysis capabilities, the paper would benefit from referencing specialized frameworks like "Open-pFind" (Chi et al., 2018) and "pNovo+" (Chi et al., 2013) that specifically address the challenges of identifying post-translational modifications.

From a methodological standpoint, the paper misses opportunities to connect with fundamental work on efficient attention mechanisms such as "Linformer" (Wang et al., 2020) and "Performer" (Choromanski et al., 2020), which pioneered approaches to reduce attention complexity from O(n²) to O(n). Additionally, when presenting its list-wise ranking approach, the paper fails to acknowledge seminal works in information retrieval like "ListNet" (Cao et al., 2007) and "ListMLE" (Xia et al., 2008) that established the mathematical foundations for list-wise ranking. These omissions don't invalidate RankNovo's contributions but limit readers' ability to understand how it builds upon established techniques across multiple disciplines.

**Experimental Designs Or Analyses:**

Limited PTM Coverage: While the datasets include some post-translational modifications, more extensive testing on heavily modified peptides would strengthen the evaluation.

**Methods And Evaluation Criteria:**

RankNovo's methods and evaluation approach are aligned with the de novo peptide sequencing problem. The list-wise reranking approach appropriately handles multiple candidate sequences with subtle differences, while the MSA representation with axial attention efficiently captures both within-peptide patterns and cross-candidate information. The mass-centric metrics (PMD and RMD) are specifically tailored to the chemistry-driven nature of peptide sequencing, and the cross-attention mechanism effectively integrates spectrum information. The evaluation methodology is thorough, using established benchmark datasets (MassIVE-KB, 9-species-V1, 9-species-V2) with standard metrics like peptide recall and amino acid precision, while incorporating diverse base models and comprehensive ablation studies to assess different components' contributions.

**Minor Limitations**

Limited PTM Coverage: While the datasets include some post-translational modifications, more extensive testing on heavily modified peptides would strengthen the evaluation.
Single Instrument Type: Testing on data from a broader range of mass spectrometry instruments would better demonstrate cross-platform generalization.

**Other Comments Or Suggestions:**

Refer to the previous section

**Other Strengths And Weaknesses:**

refer to previous section

**Questions For Authors:**

Refer to previous section

**Relation To Broader Scientific Literature:**

RankNovo integrates concepts from both proteomics and machine learning domains, building upon established research trajectories. In proteomics, it extends the evolution from traditional database search methods (SEQUEST, Mascot) to deep learning approaches for de novo peptide sequencing, following Casanovo's transformer architecture and DeepNovo's neural network foundation. However, RankNovo distinctly reframes peptide sequencing as a reranking problem rather than a generation task, adapting a strategy that has proven effective in protein structure prediction and other computational biology applications.

From a methodological perspective, RankNovo adapts several established machine learning techniques to the peptide sequencing domain. Its list-wise reranking approach parallels methods in information retrieval and NLP, while the axial attention mechanism borrows from efficient transformer variants in computer vision. The multiple sequence alignment representation repurposes traditional bioinformatics techniques, and the mass-focused metrics introduce domain-specific supervision signals. This combination addresses specific challenges in de novo peptide sequencing, particularly the discrimination between amino acids with similar masses. The paper's demonstration of zero-shot generalization connects to research on model-agnostic meta-learning, though the broader applicability of these techniques beyond peptide sequencing remains to be fully explored.

**Theoretical Claims:**

The RankNovo paper primarily focuses on empirical contributions rather than making significant theoretical claims that require formal proofs. There are no mathematical theorems, lemmas, or propositions in the paper that would necessitate rigorous proof verification. The main algorithmic contributions—the list-wise reranking approach with axial attention and the definition of PMD and RMD metrics—are presented as algorithmic descriptions with pseudocode rather than theoretical results.

The paper does briefly discuss computational efficiency claims regarding the reduction from O(n²) to O(n) complexity with axial attention, but this is a well-established result in the literature. Similarly, while the authors provide theoretical motivation for their design choices, these are presented as design rationales rather than formal claims requiring proof. In summary, the paper's contributions are predominantly empirical, focusing on the practical design, implementation, and evaluation of a novel framework for peptide sequencing.

---

> ### Author Rebuttal · Authors · 2025-04-01
>
> Thank you for your detailed comments. We address your concerns as follows:
>
> > The paper lacks validation on more challenging datasets with diverse post-translational modifications (PTMs).
>
> We sincerely appreciate your insightful observation regarding the need for validating RankNovo on datasets with more diverse post-translational modifications. You've highlighted a crucial aspect of peptide de novo sequencing that directly impacts the practical utility of our approach in comprehensive proteomics studies.
>
> ``` In our current Nine-Species-V1 benchmark, we included three classes of PTMs: Oxidation-M, Deamidation-N, and Deamidation-Q.``` The promising results of these modifications demonstrated that RankNovo can effectively enhance performance on PTM-containing spectra when such modifications are incorporated during training.
>
> To address your valuable suggestion, we conducted additional experiments on a more diverse set of biologically significant PTMs from the dataset compiled by Zolg et al. [1]. ``` Specifically, we selected three functionally important modifications: Acetylation (K), Dimethylation (K), and Phosphorylation (Y).``` Each PTM included ~62.5K spectra split 8:1:1 for training/validation/testing.
>
> Given that these new PTMs were not in the original vocabulary of our models, we performed necessary fine-tuning procedures. We combined the training and validation datasets across all three PTMs, reinitialized the embedding and final linear layers to accommodate the expanded vocabulary, and fine-tuned both the six base models and RankNovo accordingly.
>
> **Our results consistently demonstrated RankNovo's superiority:**
>
> - Acetylation (K): 5.6% improvement over the best base model (0.889 vs 0.833)
> - Dimethylation (K): 2.8% improvement (0.487 vs 0.457)
> - Phosphorylation (Y): 6.7% improvement (0.589 vs 0.522)
>
> | PTM | Casanovo | ContraNovo | ByNovo | R-Casanovo | R-ContraNovo | R-ByNovo | RankNovo |
> |---|---|---|---|---|---|---|---|
> | Acetylation (K) | 0.819 | 0.820 | 0.830 | 0.833 | 0.806 | 0.832 | 0.889 |
> | Dimethylation (K) | 0.455 | 0.459 | 0.458 | 0.458 | 0.401 | 0.457 | 0.487 |
> | Phosphorylation (Y) | 0.476 | 0.473 | 0.520 | 0.491 | 0.519 | 0.522 | 0.589 |
>
> These compelling results further validate that our deep learning reranking framework maintains its potential across a more diverse spectrum of PTMs, underscoring the robustness and broader applicability of our approach for advanced proteomics research.
>
> [1]ProteomeTools: Systematic characterization of 21 post-translational protein modifications by liquid chromatography tandem mass spectrometry (LC-MS/MS) using synthetic peptides
>
>
> > Testing on data from a broader range of mass spectrometry instruments would better demonstrate cross-platform generalization.
>
> Thank you for highlighting this limitation. While our original benchmarks (Nine-species-v1/v2) used only **Q Exactive instruments**, our training dataset (MassiveKB) **includes PSMs from various instruments, potentially enabling cross-platform generalization.**
>
> To address your concern, we note that the **PTM datasets** mentioned in our response to Q1 were collected using **Orbitrap Fusion Lumos** instruments, different from our benchmark's Q Exactive instruments. RankNovo consistently outperformed all base models on these Orbitrap Fusion Lumos datasets across all three PTMs (Acetylation-K, Dimethylation-K, and Phosphorylation (Y)).
>
> ```These results provide strong evidence of RankNovo's effectiveness beyond Q Exactive instruments, demonstrating its cross-platform generalization capabilities.``` We appreciate this suggestion which helped strengthen our evaluation.
>
>
> > The manuscript would benefit from addressing additional research directions in the literature review to better contextualize scientific contributions.
>
> Thank you for this valuable suggestion. We decided to incorporate the recommended research directions into our final manuscript as follows:
>
> - **Introduction (Section 1):**
>
> Added ensemble approaches for multiple search engines **(PepExplorer, iProphet)**
> - **Related Work - De Novo Peptide Sequencing (Section 2.1):**
>
> Incorporated PTM identification works **(Open-pFind, pNovo+)**
> - **Related Work - Candidate Reranking (Section 2.2):**
>
> Expanded with list-wise reranking algorithms **(ListNet, ListMLE)**
> - **Related Work - Axial Attention (Section 2.3):**
>
> Enhanced with efficient attention mechanisms **(Linformer, Performer)**
>
> These additions better contextualize our contributions within existing literature and help readers understand how our work bridges deep learning techniques and proteomics applications. We appreciate your feedback which has improved the manuscript's depth and quality.

---

### Decision · Program_Chairs · 2025-05-01

**Decision:**

Accept (poster)

**Comment:**

The submission presents RankNovo, a deep reranking framework designed to improve de novo peptide sequencing using multiple sequencing models. RankNovo introduces a unique reranking strategy using axial attention and introduces peptide mass deviation (PMD) and residual mass deviation (RMD) metrics, setting a new standard in sequencing benchmarks. Reviewers broadly recognised the novelty of the reranking approach and its demonstrated performance improvements, highlighting its robust generalisation to unseen models.

Review scores were 3, 3, 4, 2. The primary strengths of RankNovo include its ability to surpass previous methods in peptide recall and its successful application of axial attention. Nevertheless, reviewers had some reservations about the choice of benchmark models and the absence of testing on more challenging datasets with extensive post translation modification (PTM) coverage. The rebuttal was thorough, addressing these concerns by providing additional experimental results on diverse PTM datasets and clarifying model choices and configurations.

Reviewers engaged actively in the discussion. One raised their score after acknowledging the clarification of certain doubts, reflecting the alignment of opinion towards acceptance. The paper's comprehensive benchmarking and use of deep learning for peptide sequencing suggest meaningful practical implications in proteomics.

Given the resolution of initial concerns and the consensus on its significant contributions, I recommend accepting this paper. Its scalability and improvement across empirical tests make it a valuable addition to ICML.